# Investigating the population structure of *Moraxella catarrhalis* using a cgMLST scheme and LIN code system

Iman Yassine [1], Keith A. Jolley [2], James E. Bray [2], Melissa J. Jansen van Rensburg[1], Fadheela Patel [3], Anna E. Sheppard [1], Heather J. Zar [4], Veronica Allen [3], Lemese Ah Tow[3], Martin C. J. Maiden [2], Mark P. Nicol [3,5] & Angela B. Brueggemann [1] ✉

*Moraxella catarrhalis* is an important cause of infectious exacerbations of chronic obstructive pulmonary disease and otitis media. To investigate the population structure of *M. catarrhalis*, we developed a core-genome multi-locus sequence typing (cgMLST) scheme using 1319 core genes, and a life identification number (LIN) barcode classification system. Whole-genome analyses of nearly 2000 genomes confirmed divergent seroresistant (SR) and serosensitive (SS) *M. catarrhalis* lineages with distinct evolutionary trajectories. SR genomes are more conserved, while SS genomes exhibited greater genetic variability. Virulence gene analyses revealed lineage-specific variations in ubiquitous surface proteins (UspA1 and UspA2) and lipooligosaccharide (LOS) types. The *bro* β-lactamase, and *mcb* bacteriocin cluster, are more common in SR lineages, which suggested different selective pressures and adaptation. Here, we show that this cgMLST scheme and LIN code system provide a robust method for characterising *M. catarrhalis*, distinguish between SR and SS lineages, and offer a unified framework for population structure analyses.

The nasopharynx serves as both a protective barrier for the respiratory system and a habitat for a diverse range of bacterial species, including commensals and opportunistic pathogens[1]. One such bacterium is *Moraxella catarrhalis*, a Gram-negative, non-encapsulated diplococcus that naturally colonises the human upper respiratory tract[2,3]. While it was considered to be a commensal, it is increasingly recognised as an opportunistic pathogen responsible for respiratory infections. Carriage of *M. catarrhalis* is common, particularly in children, and it is a leading cause of otitis media, ranking just behind *Streptococcus pneumoniae* and nontypeable *Haemophilus influenzae*[3]. In adults, particularly those with chronic obstructive pulmonary disease (COPD), *M. catarrhalis* is frequently associated with acute exacerbations of

disease[2,4]. In rare but severe cases, *M. catarrhalis* can also cause invasive infections such as pneumonia, bacteraemia, and meningitis[3–5].

The pathogenicity of *M. catarrhalis* is driven by multiple virulence factors that facilitate immune evasion, adhesion, and persistence. Lipooligosaccharide (LOS), which exists in three distinct serotypes (A, B, and C) based on structural variation, plays an important role in adherence to epithelial cells and enables *M. catarrhalis* to evade immune detection and establish persistent colonisation[6,7]. Additionally, ubiquitous surface proteins A1 and A2 (UspA1 and UspA2) contribute to pathogenesis: UspA1 facilitates adhesion to host epithelial receptors and UspA2 enhances serum resistance by interacting with complement regulatory proteins[7–11]. These factors collectively

[1]Nuffield Department of Population Health, University of Oxford, Oxford, UK. [2]Department of Biology, University of Oxford, Oxford, UK. [3]Division of Medical Microbiology, University of Cape Town, Cape Town, South Africa. [4]Department of Paediatrics and Child Health, SA-MRC Unit on Child & Adolescent Health, University of Cape Town, Cape Town, South Africa. [5]Marshall Centre, School of Biomedical Sciences, The University of Western Australia, Perth, Australia. ✉e-mail: angela.brueggemann@ndph.ox.ac.uk

promote biofilm formation, bacterial aggregation, and immune evasion, enabling long-term persistence in the host[7].

The increase in antimicrobial-resistant *M. catarrhalis* is of concern. The production of β-lactamases BRO-1 and BRO-2 has led to resistance rates of 96% to β-lactam antimicrobials[12,13]. Beyond β-lactam resistance, *M. catarrhalis* has also demonstrated resistance to trimethoprim, macrolides, and tetracyclines, further complicating treatment options[14,15]. These trends underscore the need for continued genomic surveillance and novel treatment strategies.

Conventional molecular typing methods, such as restriction fragment length polymorphism analysis, 16S rRNA sequence analysis, and multilocus sequence typing (MLST), have been widely used to investigate the genetic diversity of *M. catarrhalis*[16,17]. These approaches have identified two primary genetic lineages, referred to as serosensitive (SS) and seroresistant (SR). SR strains exhibit higher virulence and resistance to complement-mediated killing, and are associated with 16S rRNA sequence type 1 (RB1). SS strains are more susceptible to immune attack, and correspond to 16S rRNA sequence types 2 and 3 (RB2 and RB3). However, these typing approaches have limitations, including limited resolution and reduced discriminatory power for closely related strains[16,17]. Advances in whole-genome sequencing approaches enable improved insights into the population structure and evolution of *M. catarrhalis* and confirm the independent evolutionary trajectories of SS and SR lineages; however, such studies have relied on a small number of genomes, limiting their ability to fully capture the genetic diversity and epidemiology of this species[18–20].

To enhance strain classification and epidemiological tracking, core genome multilocus sequence typing (cgMLST) has emerged as a more precise, standardised, and scalable alternative to single and multilocus typing methods. Unlike MLST, which analyses seven or eight housekeeping genes, cgMLST schemes analyse thousands of core genome loci, providing much higher resolution for strain differentiation. Given that cgMLST schemes capture allelic variation over many loci, each genome typically has a unique core genome sequence type (cgST), and this high resolution can complicate large-scale comparisons and analyses. Single-linkage clustering is often applied to group similar cgSTs based on defined thresholds of allelic differences; however, single-linkage clustering lacks stability, and clusters can merge when subsequent genomes bridge the designated identity thresholds. To overcome these challenges, the Life Identification Number (LIN) code, a multi-position integer barcode, has recently been introduced, which provides a stable, hierarchical classification framework. The cgMLST scheme, coupled with the LIN code system, ensures greater global comparability, reproducibility, and reliability in epidemiological studies, making these tools particularly useful for outbreak investigations, antimicrobial resistance monitoring, and evolutionary studies[21–24].

In this study, we developed a cgMLST scheme and LIN barcoding system for *M. catarrhalis* using a large genome dataset to provide a systematic and scalable framework for genomic epidemiology. These new genotyping tools provide a high-resolution, standardised approach for characterising *M. catarrhalis* population dynamics and enhanced capacity for global comparisons of isolates.

## Results

### The *M. catarrhalis* genome collection
A total of 1913 *M. catarrhalis* genomes were analysed, including previously sequenced isolates from the Drakenstein Child Health Study (DCHS) and publicly available genomes from the NCBI database. This genome collection showed high assembly contiguity, with a median N50 of 294 Kb and a median L50 of 3, reflecting a relatively well-assembled dataset with minimal fragmentation. The median genome length of 1.9 Mb (range, 1.7–2.1 Mb) was consistent with the expected genome size of *M. catarrhalis*. The *M. catarrhalis* were recovered

between 1932 and 2020 from 12 countries across six continents (Supplementary Fig. 1, Supplementary Data 1). A total of 710 ribosomal MLST (rMLST) sequence types (rSTs) and 491 MLST sequence types (STs) were represented in this collection. The STs were further grouped into 78 clonal complexes (CCs), with 219 STs classified as singletons (unclustered STs; Supplementary Data 1 and 2).

### Development and implementation of the cgMLST scheme
The cgMLST scheme was created using the development dataset of 12 complete genomes and 175 draft genomes. This initial analysis identified 3063 complete, non-redundant coding sequences, of which 1353 genes were classified as core genes present in ≥95% of genomes. To validate the scheme, the validation dataset, comprising 1726 genomes (including DCHS genomes plus assemblies from publicly available data) was used. The inclusion of these genomes resulted in a slight reduction in core genes to 1348, while the pangenome was expanded to 3714 genes, reflecting increased genetic diversity across the dataset. Paralogous genes were removed, and the preliminary cgMLST scheme was uploaded to PubMLST, where manual and automated refinement tests were performed to identify and remove problematic loci, including those with a high number of missing alleles. After these refinements, a final cgMLST v1.0 scheme, consisting of 1319 core genes, was implemented in PubMLST. As a further assessment, we re-ran chewBBACA on the complete dataset of 1913 *M. catarrhalis* genomes using the original parameters, which identified 1333 core genes, as compared to the 1353 loci initially identified from the development set of 187 genomes. A BLASTn analysis revealed that 1316 of the 1333 core genes were also in the original gene set. Ten of the 17 non-overlapping genes were present in the original pangenome of 3063 genes but did not meet the core gene threshold. These results confirmed that the original selection of core genes based upon the development dataset successfully captured the species-wide core genome and that this cgMLST v1.0 scheme was broadly representative of *M. catarrhalis* genomic diversity, based on currently available genomes.

The 1319 core genes were evenly distributed across the genome, accounting for around 75.6% of the total coding sequences (CDS) (mean CDS = 1745; Fig. 1a, Supplementary Data 1). These genes spanned a diverse range of cellular functions, and 1055 were predicted to encode proteins involved in clusters of orthologous groups (COG)-classified cellular pathways (Fig. 1b, Supplementary Data 3). Among the entire *M. catarrhalis* dataset, a total of 165,202 unique core gene alleles were assigned (Supplementary Data 4). There were 1578 unique core genome sequence types (cgSTs) in the dataset (Supplementary Data 5). Only 26 genomes were missing more than 25 core gene allele assignments (cut-off for cgST assignment), and thus, no cgST was assigned (Supplementary Data 6).

To assess the specificity of the cgMLST scheme, 159 genomes from 24 non-*M. catarrhalis* *Moraxella* species (including four highly divergent strains previously characterised as distantly related to *M. catarrhalis*[17]) were retrieved from the rMLST genome database and screened for the presence of the 1319 *M. catarrhalis* core genes. Among the non-*M. catarrhalis* genomes, between 463 (35.1%) and 1255 (95.1%) of the core genes were detected and assigned (mean, 796 genes; median, 850 genes; Supplementary Data 7). Only one genome had fewer than 25 missing alleles. This genome was identified as *Moraxella* sp. by rMLST, but its genome size and GC% were consistent with *M. catarrhalis* characteristics. Additionally, pairwise allelic mismatch analysis classified it within the SR lineage, suggesting it may represent an unspeciated *M. catarrhalis* genome.

Only *M. canis* and the four divergent strains shared more than 1200 core genes with *M. catarrhalis*; however, all non-*M. catarrhalis* genomes exhibited more than 99% pairwise allelic mismatches with *M. catarrhalis* and had more than 25 missing alleles, meaning that cgSTs were not assigned (Supplementary Data 8). This confirmed that the cgMLST scheme was specific to *M. catarrhalis* (note that these allele

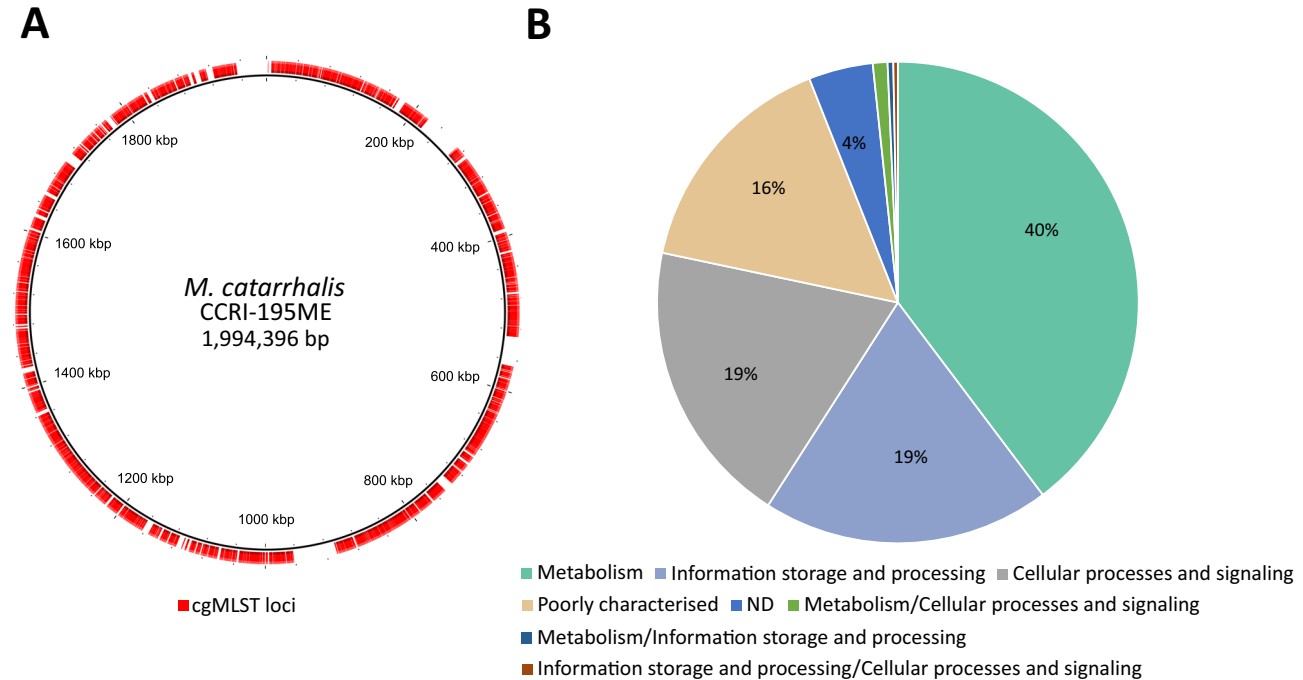

**Fig. 1 | Core gene loci used in the *M. catarrhalis* cgMLST scheme. A** Distribution of 1319 core genes (outer red ring) across the *M. catarrhalis* CCRI−195ME reference genome. **B** Cluster of Orthologous Groups (COG) categories of the 1319 core genes. Source data are provided as a Source Data file.

assignments were not included in PubMLST to avoid including non-*M. catarrhalis* alleles in the cgMLST scheme).

## Population structure and pangenome analysis of *M. catarrhalis*

The 16S rRNA analyses classified 1604 *M. catarrhalis* genomes as 16S type 1172 genomes as 16S type 2, and 16 genomes as 16S type 3[16]. Additionally, 89 genomes exhibited novel 16S types, and 32 genomes could not be assigned a 16S allele due to assembly issues at this locus, e.g., contig breaks at the end of the sequence (Supplementary Data 9). All *M. catarrhalis* genomes in the dataset consistently differed from the originally reported 16S types by a single nucleotide at position 19 (C replaced T). Since this variation was present across all genomes, it was ignored, and we continued to refer to them as 16S types 1, 2, and 3 throughout the text. To ensure accurate tracking of this variation, new alleles were assigned in PubMLST, maintaining consistency with existing 16S rRNA type designations while also capturing this genetic difference.

The cgMLST-based phylogenetic tree indicated clear separation of the *M. catarrhalis* genomes into two major clades that correlated with the 16S types. The first clade comprised 254 genomes, mainly of 16S types 2 and 3, and was designated herein as the SS clade. The second clade comprised 1659 genomes, predominantly of 16S type 1, and was designated the SR clade (Fig. 2).

To further quantify genetic variation, the percentage of allelic mismatches across 1319 core genes was computed for each genome pair and plotted. This analysis identified two major peaks: one at 82.5% mismatches, representing comparisons within the SR lineage and indicating intra-lineage diversity, and another at 99.7% mismatches, corresponding to comparisons between SR and SS genomes. Additionally, a smaller peak at 98% mismatches, representing comparisons within the SS lineage, suggested that SS is more genetically heterogeneous than SR (Fig. 3a).

To complement this analysis, we examined allele sharing across loci between the two lineages. The majority of loci (*n* = 933) shared no common alleles between SR and SS genomes, while 307 loci shared a single allele, fewer than 100 loci shared more than two, and only one locus shared nine alleles (Supplementary Fig. 2). Additionally, SS

genomes were larger than SR genomes and carried more genes (Supplementary Fig. 2). Furthermore, the pangenome analysis showed that 21 genes were uniquely present in ≥95% of SR genomes and 12 genes were exclusive to ≥95% of SS genomes. SR-specific genes were enriched in inorganic ion transport (COG P), particularly the phosphate ABC transporter system (*pstA, pstB, pstC, pstS*), which suggested enhanced phosphate uptake. Additionally, an IS200/IS605 transposase and a cold-shock protein (*capB*) suggested potential roles in genome plasticity and the stress response (Supplementary Data 10). In contrast, SS-specific genes were associated with membrane biogenesis (COG M) and secretion systems, including HlyD and TolC family proteins, associated with the type I secretion system (T1SS) (Supplementary Data 11).

Beyond these lineage-specific differences, an analysis of the *M. catarrhalis* pangenome revealed that the core genes contributed to 36% (1348/3714) of the total pangenome and that *M. catarrhalis* retains an open genome structure, with an α value < 1 (α = 0.9). While this indicated ongoing gene acquisition, the relatively stable core genome suggested that a conserved genetic framework is maintained across the population (Supplementary Fig. 3).

## LIN code classification and genetic substructure

The distribution of pairwise allelic mismatches among the 1319 cgMLST loci across all genomes was used to define the LIN code thresholds. A total of 10 thresholds were chosen to provide a range of discrimination levels. The first threshold was defined at 1290 pairwise allelic mismatches to differentiate between the SR and SS genomes. The second threshold was set at 832 allelic mismatches to flank the second peak and separate genomes with different STs that have between 63.1% and 97.8% allelic mismatches (Fig. 3b). The third threshold, set at 281 allelic mismatches, was chosen to cluster genomes with the same or different STs that have between 21.3% and 63.1% allelic mismatches (Fig. 3b). This threshold coincided with an increasing Silhouette score (St = 0.72). The fourth threshold was set at the highest Silhouette score (St = 0.75) and defined genomes with 85 pairwise allelic mismatches (Fig. 3c). Finally, high-level discrimination thresholds were set at 20, 8, 4, 2, 1, and 0 pairwise allelic mismatches.

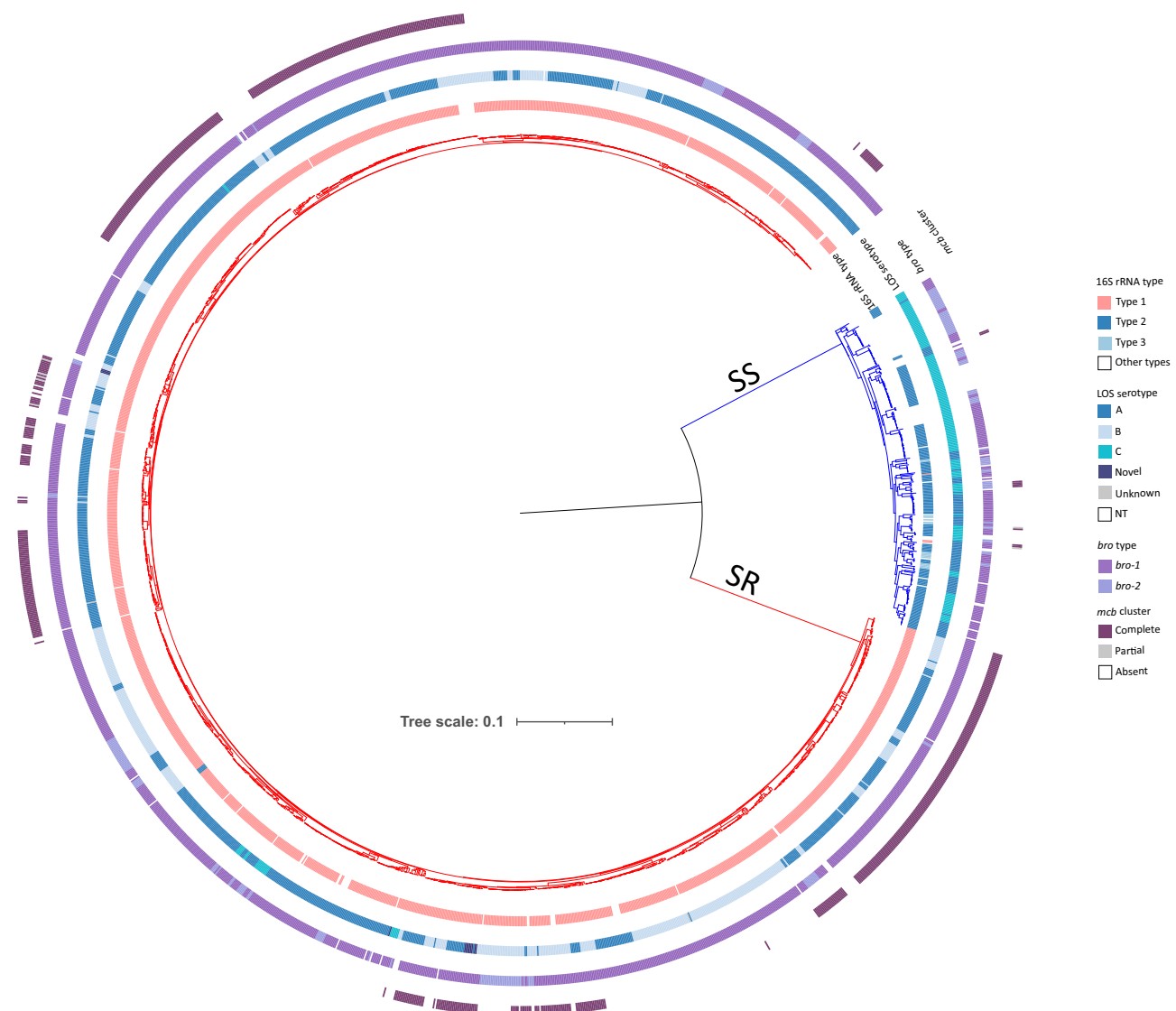

**Fig. 2 | Phylogenetic analyses of *M. catarrhalis* genomes.** Maximum-likelihood phylogenetic tree was constructed based upon a nucleotide alignment of 1319 cgMLST loci, depicting genetic relationships among 1913 *M. catarrhalis* genomes. The presence or absence of additional genes/gene regions is illustrated by the coloured rings. Note: SS, serosensitive; SR, seroresistant; NT, nontypeable; Unknown, missing data due to contig breaks. Source data are provided as a Source Data file.

Overall, there were 1887 genomes with an assigned cgST and LIN code, and 1344 unique LIN codes were defined (Supplementary Data 12). At LIN level 1 (1290 mismatches), SR and SS genomes clustered into distinct clades (Fig. 4). Within SR genomes, there were 74 unique LIN level 2 (832 mismatches), 299 LIN level 3 (281 mismatches), 526 LIN level 4 (85 mismatches), and 1154 unique LIN codes (0 mismatches). In contrast, SS genomes exhibited 54 unique LIN level 2 (832 mismatches), 84 LIN level 3 (281 mismatches), 125 LIN level 4 (85 mismatches), and 190 unique LIN codes (0 mismatches) (Fig. 4, Supplementary Data 12). The majority of genomes with matching STs had fewer than 22% allelic mismatches among the cgMLST loci; however, there were genomes with the same STs that had up to 61% allelic mismatches, while others with different STs had <5% allelic mismatches (Fig. 3b).

The Adjusted Rand Index (ARI) was used to assess the concordance between *M. catarrhalis* clusters at each LIN code threshold and their corresponding CC, ST, and rST designations. The LIN level 2 (832 mismatches) threshold showed the highest concordance with CCs (ARI = 0.65), while LIN level 3 (281 mismatches) aligned most closely with STs (ARI = 0.71), and LIN level 4 (85 mismatches) had the strongest agreement with rSTs (ARI = 0.84), reflecting the hierarchical relationship between these classification schemes (Fig. 5).

## Species classification and phylogenetic placement
Given the genomic divergence between the SR and SS lineages, we evaluated the genomes with FastANI calculation mode, which resulted in average nucleotide identity (ANI) values ranging from 95.7% to 96.2% when comparing SR and SS genomes, and confirmed that all genomes belonged to *M. catarrhalis*[25]. Furthermore, in agreement with the pairwise allelic mismatch analyses and the cgMLST phylogenetic tree, the SS lineage also exhibited a broader ANI values range (97.5-99.9%) compared to the narrow 98.8-99.9% range in SR (Fig. 6).

Since ANI values (95.7–96.2%) were close to the species delineation threshold, and to further investigate the evolutionary placement of *M. catarrhalis*, we performed rMLST and core gene phylogenetic analyses. The rMLST-based phylogenetic tree, constructed from the alignment of concatenated rMLST alleles of 25 *Moraxella* species, showed that *M. catarrhalis* formed a distinct cluster from other *Moraxella* species, which confirmed its species-level classification

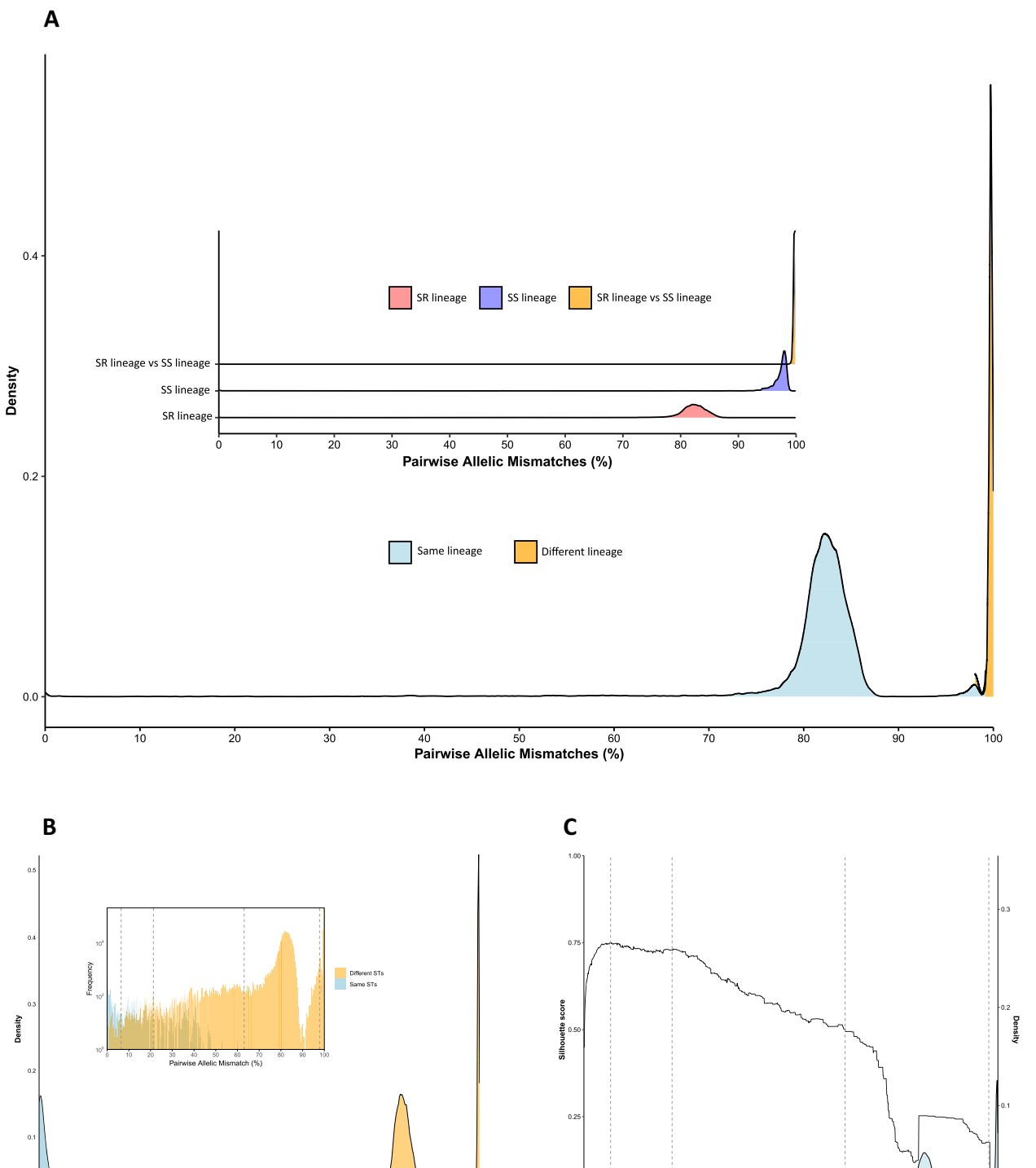

Fig. 3 | Distribution of cgMLST pairwise allelic mismatches across the *M. catarrhalis* genomes. **A** The main graph shows the overall density distribution of pairwise allelic mismatches across all genomes with assigned cgST, and the inset graph depicts specific pairwise allelic mismatch comparisons within the SR lineage, the SS lineage, and between SR and SS lineages. **B** Distribution of cgMLST pairwise allelic mismatches across the *M. catarrhalis* genomes with the same or different sequence types (STs). **C** Density distribution of pairwise allelic mismatches and corresponding Silhouette scores. Vertical lines mark the first four LIN code thresholds; the remaining six thresholds are very close together and thus not illustrated here. Source data are provided as a Source Data file.

(Fig. 7a). Consistently, the multiple sequence alignment phylogeny of 387 core genes across *M. catarrhalis* and three other *Moraxella* species reinforced this distinction and revealed intraspecies diversity not only within *M. catarrhalis*, but also within the other *Moraxella* species (Fig. 7b).

## Characterisation of *uspA*

The location of *uspA1* and *uspA2* was determined through a BLAST analysis of their sequences against the CCRI-195ME reference genome. *uspA1* is positioned between the *leuA* gene (2-isopropylmalate synthase) and a hypothetical gene encoding a threonine/serine exporter

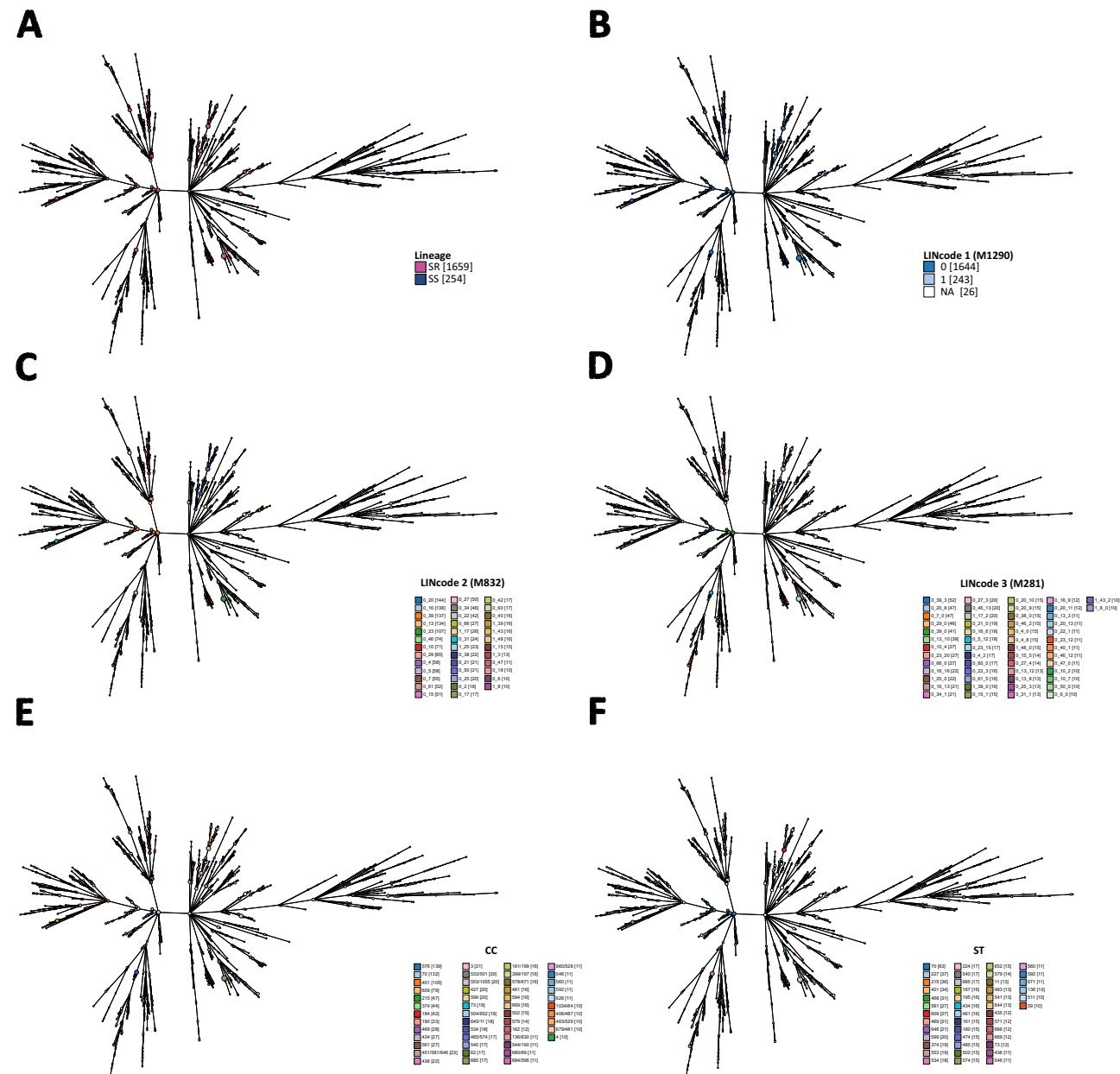

**Fig. 4 | Relationships among *M. catarrhalis* genomes depicted using GrapeTree and constructed with cgMLST allelic profiles.** Tree nodes are coloured at different levels of resolution: (**A**) SS and SR lineages; (**B**) LIN code 1 (1290 mismatches); (**C**) LIN code 2 (832 mismatches); (**D**) LIN code 3 (281 mismatches); clonal complexes (CCs); and (**F**) sequence types (STs). In panels **C**–**F**, only categories with ≥10 genomes are shown for clarity. Numbers in square brackets represent the number of genomes assigned to each LIN code, CC, or ST. NA indicates genomes for which LIN codes could not be assigned due to >25 missing cgMLST alleles. Source data are provided as a Source Data file.

family protein (referred to here as *hyp*). In 1826 out of 1913 (95.5%) genomes, both *leuA* and *hyp* were found on the same contig, whereas in 87 (4.5%) genomes, these genes were located on different contigs. While *uspA1* appeared to be ubiquitous in *M. catarrhalis*, its distribution varied between clades. In SR genomes, 96.7% (1605/1659) were classified as *uspA1*+, 2.9% (49/1659) as *uspA1*+/−, 0.2% (3/1659) as *uspA1*−, and 0.1% (2/1659) as inconclusive *uspA1*−. In SS genomes, the distribution was more variable, with 66.5% (169/254) identified as *uspA1*+, 26.8% (68/254) as *uspA1*−, 5.1% (13/254) as *uspA1*+/−, and 1.6% (4/254) as inconclusive *uspA1*− (Supplementary Data 9). The number of GGG motifs also differed between clades. In SR, repeats ranged from 2 to 8 (average = 8), whereas in SS, they varied from 11 to 66 (average = 30) (Fig. 8a, b).

The *uspA2* gene was found between *metR* (LysR family transcriptional regulator) and *tsaD* (tRNA adenosine (37)-N6-threonylcarbamoyltransferase). In 88.2% (1688/1913) of genomes, *metR* and *tsaD* were located on the same contig, while in 11.8% (225/1913), they were on different contigs. In SR genomes, 70.8% (1175/1659) were classified as *uspA2*+, 22.4% (372/1659) as *uspA2H*+, 4.7% (78/1659) as *uspA2*+/−, 2.0% (33/1659) as *uspA2H*+/−, and 0.1% (1/1659) as inconclusive *uspA2*−. In SS genomes, the classification was more uniform, with 89.0% (226/254) identified as *uspA2*+, 9.4% (24/254) as *uspA2*+/−, and 1.6% (4/254) as *uspA2H*+ (Supplementary Data 9). All SS genomes, except for four carrying *uspA2H*, displayed a distinct structure compared to the typical *uspA2* (which is characterised by the presence of LAAY-KASS-NINNY-KASS-FET motifs in the C-terminal region). Just before the CTER2

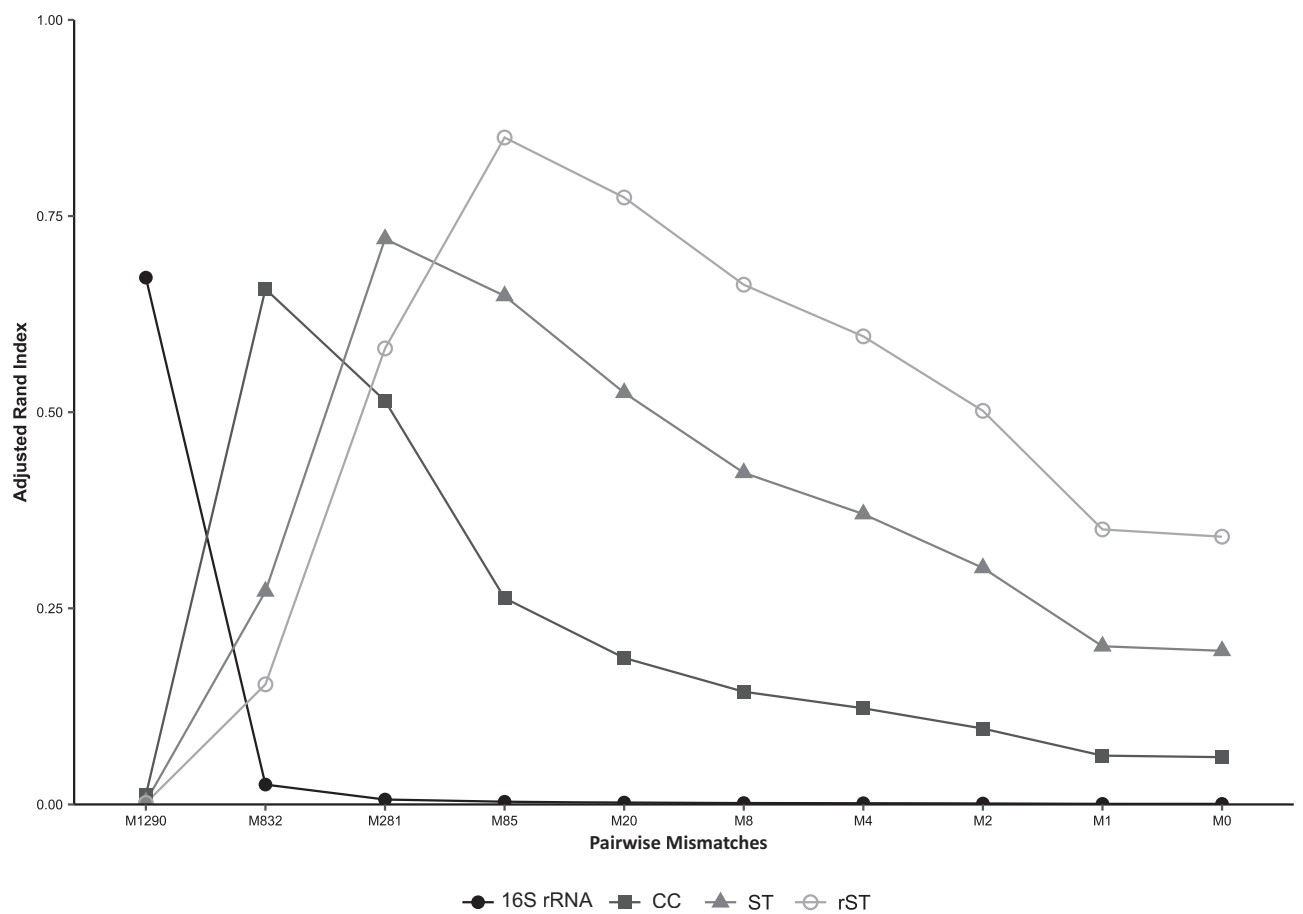

**Fig. 5 | Concordance between clusters of *M. catarrhalis* at each of the 10 LIN code thresholds and other genotyping methods.** Note: M, mismatches; CC, clonal complex; ST, sequence type; rST, ribosomal ST. Source data are provided as a Source Data file.

motif, most SS genomes exhibited a CCM/CCM-like motif−YEANN-VEDL or CCM/CCM-like motif-NINNY-KASS-FET−resembling the previously described structure of uspA2V (Fig. 8c, d).

### Characterisation of other genes of interest

LOS A was the most prevalent serotype, detected in 1168 (61 %) genomes, followed by LOS B in 531 (27.8%) and LOS C in 158 (8.3%). Fifty-six (2.9%) genomes had negative or atypical results, and among these genomes, manual BLAST analysis confirmed that 34 had a normal LOS C structure, two were incomplete due to contig breaks, and one lacked all LOS genes. Seven genomes were amplified in silico only with LOS_A primers: four had a complete LOS A structure, while three were incomplete due to contig breaks. Notably, twelve SR genomes exhibited dual amplification with LOS_A and LOS_BC primers, producing 1.9 kb and 3.3 kb amplicons. These genomes had a LOS B-like structure but carried $lgt2_A$ instead of $lgt2_{BC}$, suggesting a potential new serotype. Serotype distribution varied between SR and SS genomes. All three LOS serotypes were associated with SR genomes, whereas SS genomes had only LOS A or LOS C (Fig. 2, Supplementary Data 9).

The *bro* β-lactamase genes and the *mcb* bacteriocin cluster also showed distinct distributions between SR and SS genomes. Among SR genomes, 98% (1626/1659) harboured a *bro* gene, and *bro*-1 was the dominant variant (90.3%, 1498/1659), while 7.7% (128/1659) carried *bro*-2. The *mcb* bacteriocin cluster was present in 41% (680/1659) of the SR genomes, while 0.4% (6/1659) of genomes had a partial cluster, and 58.6% (973/1659) lacked the *mcb* bacteriocin cluster (Supplementary Data 9). Among SS genomes, 57.5% (146/254) carried *bro*-1, and 26.8% (68/254) carried *bro*-2. The *mcb* cluster was only found in 3.5% (9/254) of SS genomes, and 0.8% (2/254) of genomes

harboured a partial cluster (lacking *mcbB*), while 95.7% (243/254) lacked it entirely (Fig. 2, Supplementary Data 9).

## Discussion

The classification of *M. catarrhalis* is taxonomically complex, reflecting broader challenges in bacterial systematics. Historically, reliance on phenotypic methods led to misclassification of *M. catarrhalis* within *Micrococcus*, *Neisseria*, and *Branhamella*. The application of molecular techniques, such as DNA-DNA hybridisation and 16S rRNA sequencing, definitively placed *M. catarrhalis* within the *Moraxella* genus[3,4,26]; however, molecular and genomic studies suggested that *M. catarrhalis* was not a monophyletic species group but rather comprised two distinct genetic lineages, SR and SS[16,17,19,20]. Here, using a large *M. catarrhalis* genome dataset ($n = 1913$), robust evidence supporting this division is presented, which provides further insights into the evolutionary and functional divergence of these lineages.

The comparative genomic analyses conducted here, including cgMLST, rMLST, and ANI, confirmed that SR and SS are highly divergent yet remain within the same species group. The SS lineage revealed greater genetic diversity, with wider intra-lineage ANI values (97.5−99.9%) and increased allelic mismatches, suggesting a more heterogeneous population, while SR was more clonal and conserved. The 16S rRNA analysis, which classifies *M. catarrhalis* into RB1 (SR) and RB2/RB3 (SS) was partially supported by our findings[16,27]. While most RB1 genomes clustered with SR and RB2/RB3 with SS, we identified some RB1 genomes that grouped with SS and some RB2 genomes that clustered with SR. These findings emphasise the need to reassess older classification schemes in the context of genomic approaches that offer higher resolution and greater accuracy in delineating bacterial lineages.

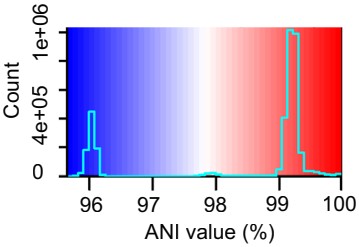

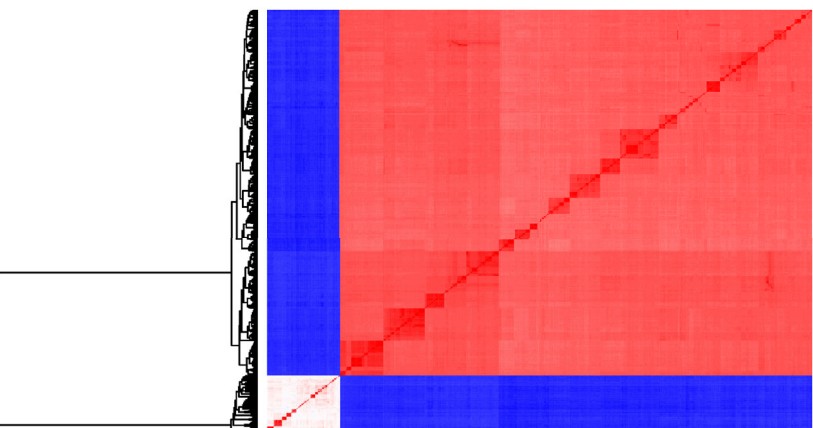

**Fig. 6 | FastANI heatmap depicting the average nucleotide identity across all 1913 *M. catarrhalis* genomes.** The main panel illustrates colour-coded pairwise ANI values, while the inset provides the colour legend. Source data are provided as a Source Data file.

Both *M. catarrhalis* lineages shared a core genome of 1348 genes. Previous studies have reported core genome sizes ranging from 1257 to 2318 genes; however, these estimates were based on a smaller number of genomes, ranging from 12 to 31 genomes[18–20]. Our findings, derived from a significantly larger dataset, provide a more comprehensive and reliable estimate of the conserved genome of *M. catarrhalis*. Moreover, the pangenome analysis revealed 3714 genes, with a moderately open pangenome structure ($\alpha = 0.9$), which suggests some degree of ongoing gene turnover at a limited rate. This is consistent with other host-associated respiratory bacteria such as *Neisseria meningitidis* and *S. pneumoniae*, whose pangenomes also remain open but expand slowly[28–30]. These findings position *M. catarrhalis* near the boundary between open and closed pangenomes, reflecting limited genomic plasticity that may support adaptation to selective pressures such as antibiotics or host immune responses, while maintaining overall genomic stability.

We implemented the LIN code system in conjunction with a cgMLST-based typing scheme to enhance strain classification in *M. catarrhalis*. Our cgMLST scheme, which examined variation across 1319 core genes, provides a standardised and high-resolution framework for tracking genomic diversity within bacterial populations. This is particularly important for *M. catarrhalis* because conventional methods such as 16S rRNA sequence analysis and MLST do not capture the full extent of intraspecies genomic diversity. For example, we observed that two genomes with the same ST were genetically divergent at the core genome level, while other genomes with different STs were nearly identical, limiting the interpretability of ST-based classifications for *M. catarrhalis*. The LIN code system builds upon cgMLST by applying a numeric-based hierarchical framework that defines genetic relationships based on allelic mismatches across loci, which avoids the instability associated with traditional clustering methods, such as single-linkage approaches that can result in unstable clusters. In contrast, LIN codes enable reproducible, scalable classification across multiple hierarchical resolutions. For example, if two isolates have LIN code 4 designations of 0_2_1_1 and 0_2_1_2, respectively, we can immediately infer that both belong to the SR lineage (shared '0' in the LIN code 1 bin), and that the genomes differ at more than 85 alleles (different LIN code 4 bin numbers) but no more than 281 alleles (identical LIN code 3 bin numbers) out of 1319 core genes, providing a quick and interpretable measure of genetic differences.

This scalable approach of cgMLST and LIN codes enables precise strain differentiation, while maintaining a degree of compatibility with existing classification schemes[21,22]. By integrating cgMLST with LIN codes, we established a robust and reproducible method for delineating *M. catarrhalis* lineages, effectively distinguishing SR and SS subpopulations with greater resolution than previous approaches. This system serves as a valuable tool for epidemiological monitoring, longitudinal surveillance, and evolutionary studies, providing deeper insights into *M. catarrhalis* population structure and its ecological adaptation. Although *M. catarrhalis* is not currently considered a major clinical threat, its high carriage rate in healthy individuals and potential role in the evolution of antimicrobial resistance underscore the importance of continued genomic surveillance[12,13,31].

Phylogenetic analysis based on ribosomal gene alignment confirmed that *M. catarrhalis* forms a distinct cluster within the *Moraxella* genus while exhibiting notable intraspecies diversity and structure. SR and SS form closely related yet genetically distinct sublineages, indicating a structured population within the species. Phylogeny based on core gene alignment reinforced this observation, revealing similar intraspecies diversity across the genus. This suggested that the presence of genetically distinct subpopulations may be a broader evolutionary trait among *Moraxella* species, potentially driven by shared selective pressures or ecological adaptations.

Beyond genetic divergence, SR and SS exhibit distinct functional profiles that align with previous studies[18,19]. The SS lineage possesses some genes related to membrane biogenesis and secretion systems,

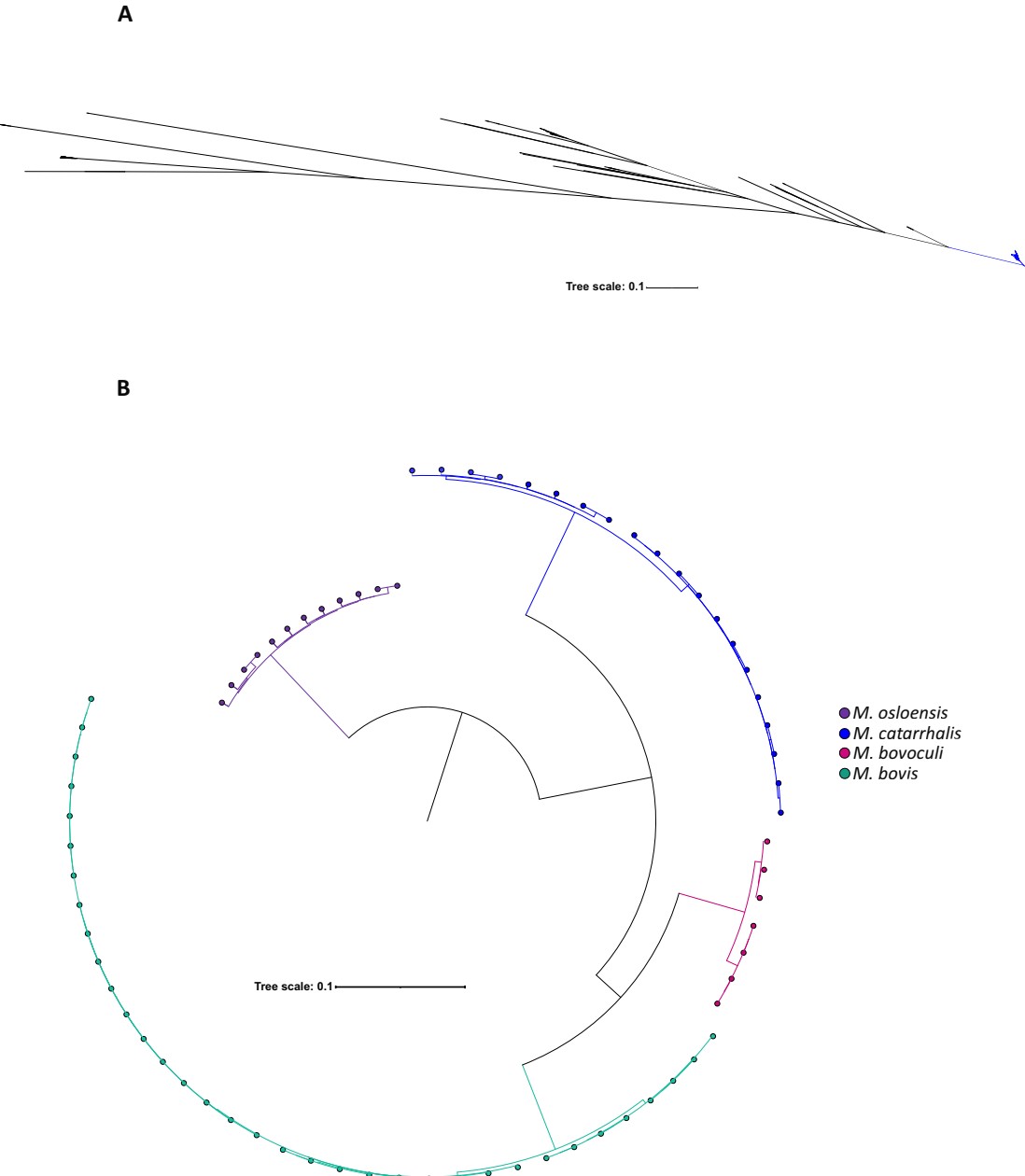

**Fig. 7 | Relationships between *M. catarrhalis* and other *Moraxella* species.**
**A** Maximum-likelihood phylogenetic tree depicting relationships between 1913 *M. catarrhalis* genomes and 155 genomes from 24 other *Moraxella* species, constructed from a nucleotide alignment of 53 rMLST loci. **B** Neighbour-joining tree constructed from a multiple sequence alignment of 387 core genes shared among four different *Moraxella* species. Source data are provided as a Source Data file.

including HlyD and TolC proteins, components of T1SS. In contrast, the SR genomes are enriched in genes associated with inorganic ion transport, particularly phosphate ABC transporters, suggesting metabolic specialisation. The presence of cold-shock proteins suggests an enhanced ability to withstand environmental fluctuations. Various studies have demonstrated that exposure to cold temperatures can significantly impact bacterial adaptation in the upper respiratory tract. Nasopharyngeal temperature can drop from 34 °C to 26 °C within minutes upon inhalation of cold air, triggering adaptive responses in resident flora[32,33]. In vitro studies have shown that a 26 °C cold shock upregulates key virulence traits in *M. catarrhalis*, including adherence to epithelial cells, iron acquisition, complement resistance, and immune evasion[34–36]. The presence of cold-shock proteins in SR genomes implies that this lineage may rely on stress-response mechanisms

for survival, potentially enhancing its ability to persist in dynamic host environments.

The SR and SS lineages exhibited differences in key virulence-associated factors. For example, the *uspA1* gene, essential for epithelial attachment and immune evasion, was nearly ubiquitous in SR. Structural variation between the lineages was also noted, e.g., in SS, *uspA1* showed greater variability in its GGG repeat region. Previous studies demonstrated that each GGG repeat forms an antiparallel β-strand, a structural feature that likely causes substantial changes in the size of the head domain[8]. These changes may, in turn, influence protein function and interactions with host cells. Similarly, *uspA2*, which plays a crucial role in serum resistance, was ubiquitous but structurally diverse. In SR, *uspA2* predominantly existed in its typical form or as the *uspA2H*

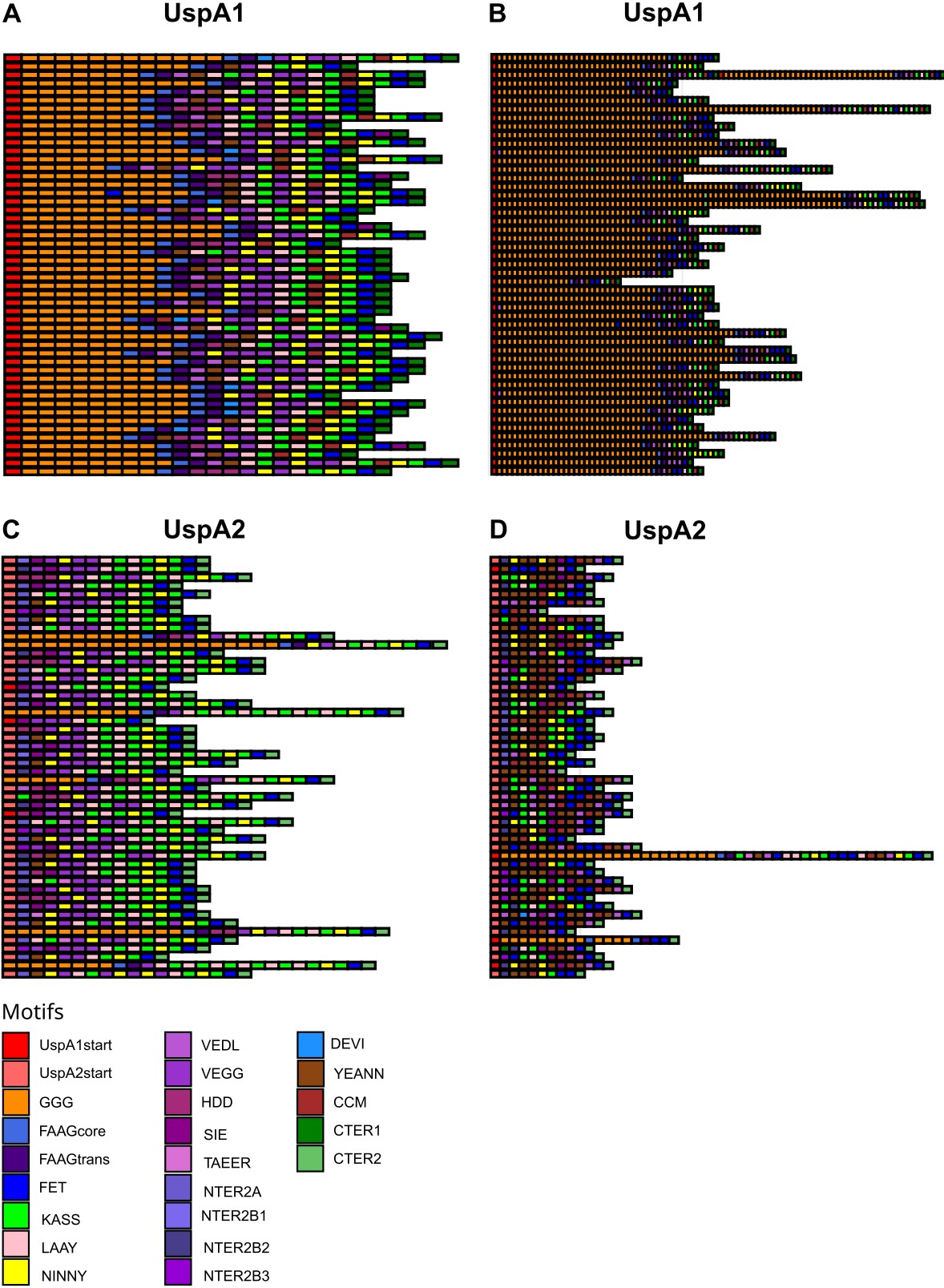

**Fig. 8 | Schematic representation of UspA in *M. catarrhalis* genomes. A** UspA1 in SR genomes. **B** UspA1 in SS genomes. **C** UspA2 in SR genomes. **D** UspA2 in SS genomes. Each coloured block represents a distinct motif, without respecting length or position. This is a representation of the 50 most common sequences. Source data are provided as a Source Data file.

variant. In contrast, SS rarely harboured *uspA2H*, and its *uspA2* more commonly resembled *uspA2V*, which featured a CCM/CCM-like motif near the C-terminal region[37]. This structural variation in UspA2V has been linked to the ability of *M. catarrhalis* to interact with Carcinoembryonic antigen-related cell adhesion molecule (CEACAMs) of the epithelial cells of the respiratory tract[11,37]. Although previous studies have functionally characterised UspA2 and shown its contribution to serum resistance via vitronectin binding, they were mainly performed using SR strains and did not evaluate the impact of sequence variation across lineages[37,38]. One study concluded that UspA2 from a SS strain did not confer serum resistance, suggesting that structural variation may impair function[39]. The role of UspA2V in serum resistance remains unexplored, and no prior studies have systematically correlated UspA diversity with SR or SS phenotypes. Overall, reported associations between genotype and phenotype in the context of virulence currently present a mixed picture, and in this context, a recent review highlighted the need for a discriminatory *M. catarrhalis* genotyping scheme[40]. Our genomic analyses reveal lineage-specific variation that can be used in the design of future experimental studies to determine the functional impact of this genetic variation on protein structure, host interactions, and pathogenicity, as well as any functional divergence between SR and SS lineages.

LOS serotype distribution also varied between the SR and SS lineages. LOS A was the predominant serotype, in agreement with previous studies[14,18]. While SR genomes contained the genes to express any one of three LOS serotypes (A, B, or C), SS genomes lacked LOS B encoding genes. Previous studies have reported inconsistent associations between LOS B and specific 16S rRNA types: while some linked LOS B to 16S rRNA type 2[18], others associated it with 16S rRNA type 1[41]; in our dataset, all LOS B genomes were SR regardless of the 16S rRNA type. This suggested that while LOS B was previously linked to a specific 16S rRNA type, it may be more closely associated with the SR lineage. Notably, a previous study reported a higher frequency of LOS type B and a lower frequency of LOS type A in respiratory isolates from adults compared to children, suggesting a possible role for LOS variation in immune evasion and/or age-related adaptation[41]. In our dataset, due to the lack of metadata for the publicly available genomes, we were unable to investigate potential associations between LOS serotype and host age. Additionally, we identified a novel LOS B-like variant in SR genomes, characterised by the presence of *lgt2A*, which suggested ongoing gene exchange and structural variation within LOS biosynthesis pathways. Further experimental studies, such as biochemical and binding assays, are needed to determine whether this novel variant can produce a distinct LOS serotype, to decipher its associated functions, and to assess whether it influences host immune recognition or serum resistance.

The *bro* β-lactamase and *mcb* bacteriocin genes were both more prevalent in SR, which supported the hypothesis that SR may be better adapted for survival in polymicrobial environments. These lineage-specific genes and differences in virulence and resistance genes raise intriguing questions about whether SR and SS occupy distinct ecological niches or exhibit different colonisation strategies within the human host.

Although our study encompasses all publicly available *M. catarrhalis* genomes to date, DCHS isolates recovered from the nasopharynx of South African children were overrepresented in this dataset. This unique DCHS dataset provides valuable insights into early-life colonisation and within-lineage diversity; however, it might not be representative of the global *M. catarrhalis* population structure or disease-associated genomic features. Nevertheless, the cgMLST scheme was based on a large set of core genes shared by all or nearly all *M. catarrhalis* but not by other *Moraxella* spp, and therefore, this study dataset was suitable for the purpose of designing a new genotyping scheme. Accommodating genetic variation when new datasets are analysed should be straightforward. Furthermore, the lack of linked clinical and demographic data for most of the publicly available genomes restricted our ability to correlate genomic variation with clinical outcomes, age, or carriage/disease status. Future studies incorporating more geographically and clinically diverse *M. catarrhalis*, including those recovered from patients with different disease manifestations, will be essential to fully characterise the biology, evolutionary dynamics, and pathogenic potential of *M. catarrhalis*.

In conclusion, our findings confirm the existence of two highly divergent yet taxonomically unified *M. catarrhalis* lineages. While SR appears more clonal, SS exhibits greater genetic diversity, and this was consistent with distinct evolutionary trajectories. The implementation of cgMLST and the LIN code system provides a robust framework for epidemiological tracking and lineage classification. Further research into the functional implications of these genetic differences, particularly in relation to host adaptation and clinical outcomes, will be important for understanding the broader significance of *M. catarrhalis* diversity.

## Methods

### Genomes from the Drakenstein Child Health Study

In this study, we used sequenced and assembled *M. catarrhalis* genomes from an intensively sampled cohort of the DCHS. The DCHS is a longitudinal birth cohort study investigating the early life determinants of health or disease, including early-life respiratory infection and associated microbiomes in South African children[42]. Mother-infant pairs were followed from birth, and a subset of infants had 2-weekly sampling of the nasopharynx for the first year of life, and subsequently at 18 and 24 months of age. For this study, we included all *M. catarrhalis* isolates from the first 137 children enroled who had near-complete sample collection and who did not develop pneumonia within the first year of life.

Total DNA of *M. catarrhalis* isolates (n = 1447) was extracted using the Maxwell(R) 16 Buccal Swab low elution volume (LEV) DNA purification kit (Promega UK) according to the manufacturer's recommendations. All extracted DNA samples were quantified using the Qubit dsDNA high sensitivity (HS) Assay Kit (Life Technologies Limited, United Kingdom). Whole genome sequencing was performed using the Illumina HiSeq X™ Ten at the Wellcome Sanger Institute, and assemblies were produced using an internal pipeline[43]. Briefly, for each sample, sequence reads were assembled into multiple draft genomes using VelvetOptimiser (v2.2.5) and Velvet (v1.2)[44]. The assembly with the highest N50 was selected for further improvement. Contigs were scaffolded using SSPACE[45], and sequence gaps were filled using GapFiller[46], ensuring high-quality draft genome assemblies.

Sequencing quality was assessed using several metrics, including contig number, assembly size, and GC content. In addition, MLST (8 gene fragments) and rMLST loci (53 protein-encoding ribosomal genes) were assessed for the presence of multiple alleles as an indicator of samples that were potentially mixed cultures[47]. Samples were classified as quality control (QC) failures and excluded from analyses if they were identified as the incorrect species, if they showed atypical features (inconclusive rMLST species identification result, unassigned rST or ST due to sequencing issues at one or more rST/ST loci), or if they had poor assembly statistics and/or features suggestive of a mixed culture (Supplementary Data 13). Where possible, QC failures were re-sequenced from a new colony pick. In total, 1429 out of 1447 genomes passed the quality control criteria and were included in the final dataset (Supplementary Data 14).

### Incorporating additional publicly available genomes

To increase the genomic diversity of our dataset, the DCHS-derived sequences were complemented with publicly available *M. catarrhalis* genomes from NCBI databases, including both assembled genomes

and raw sequencing reads available as of March 2023. All available *M. catarrhalis* genomes from the NCBI nucleotide database were retrieved, which was a total of 219 complete and draft genomes. Fourteen genomes were subsequently excluded for various reasons, such as duplicate genomes, suppression from RefSeq because of contamination, excessively short length, excessive fragmentation, or frameshifted proteins (Supplementary Fig. 4, Supplementary Data 15). In addition, the NCBI Sequence Read Archive (SRA) was searched for raw sequencing data corresponding to *M. catarrhalis*, yielding 2029 genome-derived short-read datasets. Of these, 495 genomes were not part of the DCHS and were sourced from various published and unpublished studies. From this subset, 186 genomes were identified as duplicate genomes (e.g. same biosample accession number for multiple runs, genomes already present as part of complete/draft dataset) and were excluded (Supplementary Fig. 4, Supplementary Data 14); therefore, 309 short-reads were assembled using either VelvetOptimiser (v2.2.6) or SPAdes (v3.15.3), with a range of k-mer sizes (21–133 bp)[44,48].

The quality of each genome sequence was assessed. First, assembly QC was performed using QUAST (v5.3.0) to check the genome size, N50, number of contigs, and GC content[49]. Any genome that did not meet the specified threshold was excluded (Supplementary Data 16). Second, species confirmation was performed using the rMLST species tool implemented in PubMLST[47,50]. Only genomes with 100% support for *M. catarrhalis* and a defined rMLST profile were included for further analyses. Finally, the eight locus *M. catarrhalis* MLST scheme was used to assign STs to the genomes (https://enterobase.warwick.ac.uk/species/index/mcatarrhalis)[51]. Genomes that could not be assigned an ST were excluded from further analysis to ensure the dataset consisted of well-characterised isolates. Overall, the final dataset analysed included 1913 genomes: (i) 187 out of 205 complete/draft genomes; (ii) 297 out of 309 assemblies from publicly available data; and (iii) 1429 out of 1447 genomes from the DCHS dataset.

### Development of cgMLST scheme

To develop the cgMLST scheme, the genomes were split into two non-overlapping datasets: (i) the 'development' dataset, which was used to construct the cgMLST scheme and consisted of the 187 complete/draft genomes from NCBI; and (ii) the 'validation' dataset, which was comprised of 1726 draft genomes (297 assembled genomes and 1429 DCHS genomes).

The chewBBACA software suite (v3.3.1) was employed for cgMLST scheme development, using default parameters[52]. The CreateSchema module identified 3063 complete, non-redundant coding sequences across all genomes, and alleles were defined for each gene using AlleleCall. To remove potential paralogous genes, 51 genes were filtered out using the RemoveGenes option. Annotations for each gene were retrieved via UniprotFinder. Next, the ExtractCgMLST module identified 1353 core genes (i.e., genes present in ≥95% of genomes), and genes with alleles that were 20% larger or smaller than the modal length distribution were excluded.

The validation dataset was used to refine and assess the robustness of the scheme. The AlleleCall module was run using the 1353 loci, and the cgMLST scheme was redefined using the JoinProfiles and ExtractCgMLST modules. This process identified 1348 core genes that were present in ≥95% of the genomes. To further refine the scheme, a manual BLAST search was performed using blastn (v2.13.0) to remove any putative paralogous genes[53]. Genes that shared >80% sequence identity and an overlap of at least 100 bp with another gene were excluded. After this filtering step, a total of 1344 core genes were retained for further analyses.

To evaluate the representativeness and stability of the initial scheme, we ran the same chewBBACA schema-building workflow (using the original parameters) on the full dataset of 1913 *M. catarrhalis* genomes, which included all draft and complete assemblies[52]. A total of 1333 core genes were identified across the full dataset. To assess overlap, a blastn (v2.13.0) comparison was performed between the 1333 newly identified core genes and the original 1353 loci[53]. Gene matches were defined as having >80% sequence identity and >80% alignment length.

### Implementing the cgMLST scheme into PubMLST

All genome assemblies used in this study were uploaded to the PubMLST *Moraxella* database (https://pubmlst.org/bigsdb?db=pubmlst_moraxella_isolates)[50]. To establish a provisional cgMLST scheme, 1343 core genes from the CCRI-195ME complete genome of *M. catarrhalis* (CP018059.1) were deposited in PubMLST (note that one core gene was absent in this reference genome). Alleles were automatically assigned to the selected core gene loci across all genomes using the BIGSdb 'sequence tagging' function, applying the following parameters: complete coding sequences with 90% minimum identity, 90% minimum alignment, blastn word size of 20, and stop codons within 5% length difference of match[50].

To improve the accuracy and reproducibility of the scheme, manual curation was performed for 124 genes that were assigned alleles in <97% of the genomes. Twenty-four of these genes contained a high proportion of missing alleles or invalid alleles (i.e., lacking a start or stop codon, or having internal stop codons), and these genes were excluded from the cgMLST scheme. Additionally, start codon positions for some loci were manually adjusted where the original reference annotation was inconsistent across the total dataset. In these cases, an alternative, conserved start codon was identified and used to redefine the gene boundaries.

After curation and filtering, a total of 1319 core genes were retained in the final cgMLST scheme, and each gene was present in >95% of genomes. A cgST was assigned to each genome that had allele designations for >98% of the core genes (i.e., a maximum of 25 missing alleles). To further classify genetic relationships, a LIN code system was also applied[21,22,24]. A LIN code is composed of multiple bins (positions), each corresponding to a specific cgMLST allelic mismatch threshold. Moving from left to right along the code, the bins represent decreasing allele mismatch (i.e., increasing similarity). We defined a *M. catarrhalis* LIN code system that consisted of 10 thresholds: the first four thresholds of 1290, 832, 281, and 85 pairwise allelic mismatches defined the deepest hierarchical levels of relatedness; and the last six thresholds of 20, 8, 4, 2, 1, and 0 pairwise allelic mismatches provided high-resolution epidemiological classification.

Future cgSTs and LIN codes will be assigned via the PubMLST *Moraxella* database. Briefly, when a new genome is added to the *Moraxella* database, the bacterial species is predicted using rMLST. If the genome is identified as *M. catarrhalis*, a human curator uploads the genome and provenance data, and the BIGSdb automated curation tools annotate the 1319 cgMLST loci. Previously defined alleles are assigned by the autotagger, then any new alleles are defined if the sequence meets the criteria of representing a complete coding sequence with ≥97% sequence identity and 99% sequence alignment length to an exemplar allele (i.e. a subset of the total number of alleles defined for each locus such that all known alleles are within 3% sequence identity of an exemplar of the same length). Any alleles not automatically assigned are manually reviewed and assigned by a curator. A cgST is then assigned automatically if the genome has ≤25 missing core gene alleles. LIN codes are assigned for any new cgSTs[21,22]. LIN code assignments are made using the curated cgMLST data, comparing the new data to all existing cgMLST and LIN code data. A LIN code is assigned to a cgST by matching it to all known LIN-encoded cgSTs: if an exact match is found, then the same LIN code is assigned, but if not, then a novel LIN code is created and the bin for the corresponding threshold is incremented.

## Phylogenetic and genomic analyses

Using the total dataset, a concatenated sequence of all 1319 core genes was generated for each genome. The concatenated sequences were aligned using MAFFT (v7.525), and missing alleles were treated as gaps[54]. Snp-sites (v2.5.1) was applied to extract polymorphic sites (total alignment length 175,014 bp)[55]. Phylogenetic trees were constructed to analyse genetic relationships among the *M. catarrhalis* genomes. An initial tree was built using IQ-TREE2 (v2.2.6) with the ModelFinder Plus option, ascertainment bias correction (-m MFP + ASC), and 1000 bootstrap replicates[56]. The final tree was then constructed based on a recombination-free alignment using ClonalFrameML (v1.13)[57]. The tree was midpoint-rooted and visualised with interactive tree of life (iTOL) (v7)[58]. A minimum spanning tree was constructed using GrapeTree based on the cgST allelic profiles[59].

The functional classification of the core genes was performed using EggNOG-mapper (v2.1.12)[60,61]. Each gene was assigned to a COG category, which was further grouped according to the Kyoto Encyclopaedia of Genes and Genomes (KEGG) BRITE functional hierarchy system[62]. To assess overall genomic similarity, the ANI was computed for all genomes using ANIclustermap (v1.2) with FastANI calculation mode and default parameters (https://github.com/moshi4/ANIclustermap)[63].

## Pairwise allelic mismatch analysis and comparison of clustering methods

Pairwise allelic mismatches among the total dataset were calculated to assess the genetic diversity and population structure of *M. catarrhalis*. For each genome pair, the number and percentage of loci with differing allele assignments were determined. The Silhouette index (St) was calculated for the full range of pairwise allelic mismatches among the core genes to evaluate the optimal clustering resolution. Pairwise allelic mismatches and the St were assessed across all genomes using MSTclust (v0.21b) with default parameters (https://gitlab.pasteur.fr/GIPhy/MSTclust).

An ARI was calculated using the R mclust (v6.1.1) package to evaluate the concordance between LIN code thresholds and clusters defined based on rST, ST, and CC classifications[64]. CCs based on the eight-locus MLST scheme were determined using the global eBURST algorithm implemented in PHYLOViZ (v2.00)[65]. CCs were defined at the single locus variant (SLV) level and named according to the predicted founder sequence type(s).

## Validation of the cgMLST scheme across *Moraxella* species

To assess the specificity of the *M. catarrhalis* cgMLST scheme, genome assemblies from other *Moraxella* species were analysed to determine whether the 1319 core loci could be assigned alleles (i.e., they were shared between *Moraxella* species and highly similar at a nucleotide sequence level). There were 165 non-*M. catarrhalis* genomes in the rMLST genome database (https://pubmlst.org/species-id) as of January 2025, and all those with an assigned rST (n = 155) were downloaded (Supplementary Data 17)[47,50]. Additionally, four highly divergent strains that were previously characterised as distantly related to *M. catarrhalis* were included[17]. Although these strains did not meet the quality control criteria and were excluded from our final dataset (high GC%, low rMLST allelic support value), they were included here to examine gene content. The cgMLST scheme was retrieved from PubMLST and applied locally using chewBBACA with the PrepExternalSchema, AlleleCall, and AlleleCallEvaluator modules[52]. Alleles were assigned across the 1319 loci where possible, and the resulting cgST profiles were analysed to determine the number of shared core loci and whether any non-*M. catarrhalis* genomes qualified for a cgST (i.e., 25 or fewer missing alleles).

## Pan-genome post-analysis

The gene presence/absence matrix of the *M. catarrhalis* pangenome generated by chewBBACA was used to assess the openness of the pangenome. Pangenome openness was determined using Heaps' law, where the pangenome is considered open if α ≤ 1 and closed if α > 1, as calculated with the R micropan (v.2.1) package[66]. Pangenome and core-genome accumulation curves were estimated from the same matrix using PanGP (v1.0)[67], employing the 'totally random' sampling method with ten sampling repeats.

## Phylogenetic placement of *M. catarrhalis* among *Moraxella* species

To assess the phylogenetic relationship of *M. catarrhalis* relative to other *Moraxella* species, a concatenated cgMLST core gene alignment was generated for *M. catarrhalis* and three other *Moraxella* species. The NCBI genome database was queried for all available complete genomes of non-*M. catarrhalis Moraxella* species as of March 2023. Only *Moraxella* species with at least seven complete genomes were included in the analysis, resulting in the selection of *M. bovis* (n = 35), *M. osloensis* (n = 12), and *M. bovoculi* (n = 7). A subset of 20 *M. catarrhalis* genomes, including both SR and SS, were included in the analysis. For SR, all complete genomes were selected, while for SS, eight draft genomes were randomly chosen (Supplementary Data 18). The chewBBACA pipeline was used to construct the cgMLST alignment and a core-genome neighbour-joining tree by running the CreateSchema, AlleleCall, SchemaEvaluator, and AlleleCallEvaluator modules[52].

The rMLST genome database in PubMLST was queried to construct an rMLST tree, including all *Moraxella* genomes with an assigned rST (n = 155) in the database (Supplementary Data 17) and all *M. catarrhalis* genomes in our study dataset (n = 1913)[47,50]. Briefly, the sequences of the 53 ribosomal loci for each genome were extracted, concatenated, and aligned using MAFFT (v7.525)[54]. SNP-sites (v2.5.1) was applied to identify polymorphic positions, and a maximum-likelihood tree was constructed using IQ-TREE2 (v2.2.6) with default parameters, but with -m MFP + ASC for model selection[55,56].

## Identification and classification of *uspA1*, *uspA2* and *uspA2H*

Since *uspA* genes contain multiple repeats and common motifs their sequences are highly variable; therefore, BLAST searches and precise identification of these genes is difficult[7–10]. To address this, we first determined the locations of *uspA1* and *uspA2* in *M. catarrhalis* CCRI-195ME using blastn (v2.13.0)[53], which revealed that *uspA1* is located between *leuA* (2-isopropylmalate synthase) and a hypothetical gene encoding a threonine/serine exporter family protein, whereas *uspA2* is positioned between *metR* (LysR family transcriptional regulator) and *tsaD* (tRNA adenosine (37)-N6-threonylcarbamoyltransferase). To locate and extract these genes in other *M. catarrhalis* genomes, BLAST searches were performed using the flanking genes, *leuA-hyp* (for *uspA1*) and *metR-tsaD* (for *uspA2*). Once identified, the intergenic regions were extracted for further analysis using the following approach: if the flanking genes were located on the same contig, the entire intergenic region was extracted and analysed. If they were located on different contigs, the available sequences extending from the gene to the contig boundary were extracted.

For all extracted sequences, open reading frames (ORFs) were identified using ORF Finder (v1.8)[68], translated, and analysed for conserved motifs via MAST (MEME Suite v5.5.6, default parameters)[69]. Genes were classified based on the presence of these motifs. When the flanking genes were on the same contig, *uspA1* was classified as *uspA1*+ if it had a GGG motif at the start and a CTER1 motif at the end. Similarly, *uspA2* was classified as *uspA2*+ if it lacked GGG motifs and FAAG repeats but contained CTER2, while *uspA2H* was classified as *uspA2H*+ if it contained GGG motifs, FAAG core and transition repeats, and ended with the CTER2 motif. If these structures were absent, the genes were classified as *uspA1*−, *uspA2*−, or *uspA2H*−. When the flanking genes were on different contigs, the genome was classified as *uspA1*+,

$uspA2^+$, or $uspA2H^+$ if both defining motifs were present on the same contig. If only partial motifs were detected, the gene was designated as $uspA1^{+/-}$, $uspA2^{+/-}$, $uspA2H^{+/-}$, or $uspA2$-$2H^{+/-}$, depending on the case. When no motifs were found, the gene was marked as inconclusive $uspA1^-$ or $uspA2^-$, indicating uncertainty due to potential sequencing errors, assembly fragmentation, or possible gene absence. The motif sequences used for MAST input are given in Supplementary Data 19.

### Detection of *bro* and *mcb* genes

The presence of the *bro* genes was identified using ResFinder (v4.5.0) with default parameters on assembled genomes[70]. For the detection of bacteriocin *mcbABCI* genes, blastn (v2.13.0) was used with a threshold of 70% sequence identity and 50% alignment coverage against reference sequences (Supplementary Data 20)[53].

### LOS serotyping and 16s rRNA typing

In silico LOS serotyping was performed via PubMLST, using the in silico PCR tool and allowing a primer mismatch tolerance of three nucleotides on assembled genomes[50]. Published primers were used to distinguish between serotypes A, B, and C, which differ in their LOS gene composition[71]. Serotype classification was based on amplicon sizes: LOS A was identified by amplification with LOS_A and LOS_BC primers (1.9 kbp and 4.3 kbp), LOS B with LOS_BC primers (3.3 kbp), and LOS C with LOS_BC primers (4.3 kbp). Atypical or negative PCR results were further examined using manual blastn (v2.13.0) analysis against *lgt1*, *lgt2*$_A$, *lgt3*, *lgt4*, and *lgt2*$_{BC}$ to confirm LOS gene presence (Supplementary Data 20)[53].

16S rRNA sequence typing was performed in silico to classify genomes based on previously published 16S rRNA types, which differ by single-nucleotide variants (SNVs) within the first 540 nucleotides of the gene (Supplementary Data 20)[16]. A locus named 16S_rRNA was defined in PubMLST, and all genomes were automatically scanned using blastn with 90% minimum identity and 90% length coverage[50,53]. If a genome had a 100% sequence identity and a full-length alignment match to a known allele, it was assigned that allele. If any differences were detected, a new allele was designated, which identified novel 16 s rRNA types.

### Reporting summary

Further information on research design is available in the Nature Portfolio Reporting Summary linked to this article.

### Data availability

All genome sequences used in this study are publicly available in PubMLST (https://pubmlst.org/organisms/moraxella-spp) as well as the NCBI (SRA or nucleotide) databases. Short-read sequences of the DCHS samples have been deposited in the European Nucleotide Archive under study number PRJEB25371. Genome accession numbers are provided for each isolate in PubMLST and listed in Supplementary Data 1. Source data are provided with this paper.

### Code availability

The code used to generate the figures is adapted from: https://github.com/brueggemann-lab/pgl_cgmlst_2024.

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

## Acknowledgements

The authors are grateful to Stephen Bentley, Danielle Walker, Siobhan Austin-Guest, Sara Sjunnebo, and Chrispin Chaguza at the Sanger Institute for the high-quality genome sequencing of the DCHS *M. catarrhalis*. This study was funded by a Wellcome Trust Investigator Award to ABB (grant number 206394/Z/17/Z) and a Wellcome Trust Biomedical Resource Grant to MJCM, ABB, and KAJ (grant number 218205/Z/19/Z). The Drakenstein study, led by HJZ and MPN, was funded by the Gates Foundation (OPP1017641 and OPP1017579), the NIH H3Africa (1U01AI110466-01A1), the South African Medical Research Council, and the Wellcome Trust (221372/Z/20/Z).

## Author contributions

Conceptualisation: I.Y., K.A.J., and A.B.B. Microbiology and/or DNA extractions: F.P., V.A., L.A.T., M.P.N., and A.B.B. DCHS Principal Investigators: H.J.Z. and M.P.N. Genome assembly: I.Y. and J.E.B. Data acquisition and/or curation: I.Y., K.A.J., J.E.B., M.J.J.vR., F.P., A.S., and V.A. Data analyses: I.Y., K.A.J., J.E.B., M.J.J.vR., F.P., A.S., and A.B.B. Data visualisation: I.Y. and A.B.B. PubMLST platform and software development: J.E.B., K.A.J., M.C.J.M. and A.B.B. PubMLST funding and infrastructure: K.A.J., M.C.J.M. and A.B.B. Writing of first draft: I.Y. and A.B.B. All authors contributed to the final version of the manuscript.

## Competing interests

The authors declare no competing interests.
