## [Transparent Peer Review file · Nature Communications]

Investigating the population structure of *Moraxella catarrhalis* using a cgMLST scheme and LIN code system

Corresponding Author: Professor Angela B Brueggemann

Version 0:

Reviewer comments:

Reviewer #1

(Remarks to the Author)

This manuscript presents a comprehensive genomic analysis of *Moraxella catarrhalis* using core genome multilocus sequence typing (cgMLST) and a life identification number (LIN) code classification system. The authors analyze nearly 2,000 genomes to investigate the population structure of this respiratory pathogen, confirming the existence of seroresistant (SR) and serosensitive (SS) lineages with distinct evolutionary trajectories. While the study offers valuable insights into *M. catarrhalis* population structure and implements useful typing schemes, it suffers from significant methodological shortcomings, lacks novelty in its main findings, presents some contradictory results, and fails to sufficiently contextualize the implications of the observed genomic differences.

The authors have assembled an impressive dataset of nearly 2,000 genomes to investigate *M. catarrhalis* population structure, confirming the previously described seroresistant (SR) and serosensitive (SS) lineages. However, I'm concerned about the heavy sampling bias in the dataset. With 1,429 of 1,913 genomes coming from the Drakenstein Child Health Study, the analysis is overwhelmingly weighted toward a single geographic region and demographic group. This bias isn't adequately acknowledged or addressed, raising questions about how representative these findings are of global *M. catarrhalis* diversity.

The methodology for developing the cgMLST scheme is problematic. Building the scheme from only 187 genomes and then applying it to a vastly larger validation dataset risks missing important genetic diversity. Why not use a more representative sampling strategy or an iterative approach that would better capture the full genetic spectrum?

The introduction of the LIN code system could be valuable, but the manuscript doesn't make a compelling case for how this improves upon existing typing methods or provide clear guidance for implementation. The thresholds chosen for LIN levels seem somewhat arbitrary, and I'd like to see stronger justification.

What's missing most from this work is a meaningful exploration of the functional significance of the genomic differences identified between SR and SS lineages. Yes, differences in virulence-associated genes and antimicrobial resistance determinants are noted, but without connecting these to clinical outcomes or ecological adaptations, it's hard to appreciate their biological relevance. The analysis of virulence factors, particularly UspA1 and UspA2 structural variations, feels superficial without exploration of how these variations might affect protein function.

Finally, the pangenome analysis lacks depth. The authors report an α value of 0.9 but don't sufficiently contextualize what this means for *M. catarrhalis* evolutionary dynamics or how it compares to related species.

(Remarks on code availability)

Is this the right repo at all? The description mentions: Pneumococcal Genome Library cgMLST Typing Scheme.

Reviewer #2

(Remarks to the Author)

1. Comparative Advantages of cgMLST and LIN Code Typing Methods:

Have other researchers published similar studies utilizing the cgMLST and LIN code typing methodologies? If so, what are the distinctive advantages of your approach compared to existing methods?

2. Hierarchical Interpretation of LIN Code Classification:

The study establishes LIN code classifications based on allele mismatch rates and correlates LIN code 1, 2, and 3 with ribotype, clonal complex (CC), and sequence type (ST), respectively. While this demonstrates consistency with established typing systems in resolution, further clarification is needed regarding the phylogenetic or taxonomic hierarchy represented by LIN codes. Specifically, what deeper biological or evolutionary implications (e.g., population structure, ancestral relationships, or functional divergence) do these hierarchical levels reflect beyond mere technical concordance?

3. Clinical and Functional Validation of Lineage-Specific Traits:

(i) The manuscript highlights differences in virulence and antibiotic resistance profiles between the two lineages. Are there publicly available clinical datasets (e.g., disease severity, treatment outcomes) that support associations between these lineages and observed phenotypic variations?

(ii) Regarding the proposed virulence determinants (UspA structural variation and novel LOS serotypes), the authors acknowledge the need for experimental validation. Could the authors elaborate on specific hypotheses or mechanistic studies (e.g., animal models, gene knockout/complementation, or phenotypic assays) planned to confirm the functional significance of these genetic differences?

(Remarks on code availability)

Reviewer #3

(Remarks to the Author)

Dear Authors,

General comment: Congratulations on your research! The topic is of great importance, especially in this context of modern investigation the population structure of *M. catarrhalis* in relation to virulence.

THE MANUSCRIPT „Understanding the population structure of *Moraxella catarrhalis* using core genome multilocus sequence typing (cgMLST) and a life identification number (LIN) code classification system” is well written in accordance with the guidelines of the journal. The title is accurate and relevant. The paper has an informative abstract. All figures and tables are necessary and understandable. The names of organisms are used appropriately. The data were systematically analyzed. The study presents relevant information for the field of two divergent SR and SS *M. catarrhalis* lineages with distinct evolutionary trajectories using whole-genome analyses of nearly 2 000 genomes from 12 countries over a long period of time. The discussion and conclusions are correct and reflect the evidence provided in the paper with a few necessary additions. The references were accurate. This work supports the discovery with the focus on the population genetics and its relationship with the virulence.

I just have two questions.

The authors said that “LOS serotype distribution varied between the SR and SS lineages”. I would like to ask the authors if they have made a comparison between their results obtained with SS an SR *M. catarrhalis* and association between more virulent strains and LOS B?

And also following the discovery of ST and CC by cgMLST and evidence that SS genomes were larger than SR genomes and carry more genes, can it be said whether there are any ST or CC associated with higher virulence?

I recommend publishing it in Journal “Nature Communications” with minor revision.

(Remarks on code availability)

Dear Authors,

General comment: Congratulations on your research! The topic is of great importance, especially in this context of modern investigation the population structure of *M. catarrhalis* in relation to virulence.

THE MANUSCRIPT „Understanding the population structure of *Moraxella catarrhalis* using core genome multilocus sequence typing (cgMLST) and a life identification number (LIN) code classification system” is well written in accordance with the guidelines of the journal. The title is accurate and relevant. The paper has an informative abstract. All figures and tables are necessary and understandable. The names of organisms are used appropriately. The data were systematically analyzed. The study presents relevant information for the field of two divergent SR and SS *M. catarrhalis* lineages with distinct evolutionary trajectories using whole-genome analyses of nearly 2 000 genomes from 12 countries over a long period of time. The discussion and conclusions are correct and reflect the evidence provided in the paper with a few necessary additions. The references were accurate. This work supports the discovery with the focus on the population genetics and its relationship with the virulence.

I just have two questions.

The authors said that “LOS serotype distribution varied between the SR and SS lineages”. I would like to ask the authors if

they have made a comparison between their results obtained with SS an SR M. catarrhalis and association between more virulent strains and LOS B?

And also following the discovery of ST and CC by cgMLST and evidence that SS genomes were larger than SR genomes and carry more genes, can it be said whether there are any ST or CC associated with higher virulence?

I recommend publishing it in Journal "Nature Communications" with minor revision.

Version 1:

Reviewer comments:

Reviewer #1

(Remarks to the Author)

The authors have addressed all my comments, and the manuscript is now improved.

Reviewer #1

This manuscript presents a comprehensive genomic analysis of *Moraxella catarrhalis* using core genome multilocus sequence typing (cgMLST) and a life identification number (LIN) code classification system. The authors analyze nearly 2,000 genomes to investigate the population structure of this respiratory pathogen, confirming the existence of seroresistant (SR) and serosensitive (SS) lineages with distinct evolutionary trajectories. While the study offers valuable insights into *M. catarrhalis* population structure and implements useful typing schemes, it suffers from significant methodological shortcomings, lacks novelty in its main findings, presents some contradictory results, and fails to sufficiently contextualize the implications of the observed genomic differences.

We thank the reviewer for the positive comments and the opportunity to clarify other points that were not clear enough in our first version of the manuscript.

Comment	Response	Location
1. The authors have assembled an impressive dataset of nearly 2,000 genomes to investigate M. catarrhalis population structure, confirming the previously described seroresistant (SR) and serosensitive (SS) lineages.	We fully agree with the reviewer that it is essential to consider biases that may affect the interpretation and generalisability of this work, and we have edited the original text to emphasise this point. Please note that we included all M catarrhalis genomes that were available in the public domain and passed quality control assessments in the development of this genotyping scheme, i.e. 187 out of 219 possible	lines 393-405

However, I'm concerned about the heavy sampling bias in the dataset. With 1,429 of 1,913 genomes coming from the Drakenstein Child Health Study, the analysis is overwhelmingly weighted toward a single geographic region and demographic group. This bias isn't adequately acknowledged or addressed, raising questions about how representative these findings are of global M. catarrhalis diversity.	genomes from NCBI and 297 out of 495 possible genomes from the SRA that were not part of the DCHS. Therefore, while the dataset is skewed toward carriage isolates from the DCHS, the scheme was based on all publicly available genome data, in addition to our as yet unpublished DCHS dataset. The cgMLST scheme was based on a large set of 1,319 core genes shared by all or nearly all M. catarrhalis but not by other Moraxella spp, and therefore accommodating genetic variation when new datasets are analysed should be straightforward. This study dataset was suitable for the purpose of designing of a new genotyping scheme, and these new genotyping and classification tools are important steps forward in deciphering the biology of M. catarrhalis. We have added a new paragraph to the Discussion, highlighting the implications of this bias and encouraging future genome and metadata submissions.	
2. The methodology for developing the cgMLST scheme is problematic. Building the scheme from only 187 genomes and then applying it to a vastly larger validation dataset risks missing important genetic diversity. Why not use a more representative sampling strategy or an iterative approach that would better capture the full genetic spectrum?	This is also addressed in the response above: we worked with all publicly available genomes and selected the 187 genomes that were in NCBI as complete or high-quality draft genomes. Please also note that these analyses have revealed previously undescribed diversity among the 16S rRNA type (types 1, 2 and 3 plus novel 16S types), LOS serotype (A, B and C plus novel types), bro-1 and bro-2, differentiation between SS and SR, and variation in the mcb cluster. To address the reviewer's concern and further evaluate the robustness of the cgMLST scheme, we re-ran the cgMLST schema-building pipeline using ChewBBACA on all 1,913 genomes, applying the same parameters. This analysis identified 1,333 core genes, as compared to 1,353 in the original 187 genome subset. Furthermore, a BLASTn analysis revealed that 1,316 of the 1,333 core genes in the dataset of 1,913 genomes were included in the original scheme. Ten of the remaining 17 genes were present in the original pangenome of 3,063 genes. These new results confirmed that our original core gene set captures the vast majority of conserved loci across the species, based on all currently available M. catarrhalis genomes. These new validation results have been included in the revised manuscript.	lines 112-120, 490-495

3. The introduction of the LIN code system could be valuable, but the manuscript doesn't make a compelling case for how this improves upon existing typing methods or provide clear guidance for implementation. The thresholds chosen for LIN levels seem somewhat arbitrary, and I'd like to see stronger justification.	We have revised the manuscript to provide a more detailed explanation of the LIN code system and its advantages over conventional bacterial typing methods. Traditional methods such as MLST lack sufficient resolution for fine-scale analysis. This is unsurprising, given that the MLST schemes characterise internal fragments of seven or eight housekeeping genes, whereas cgMLST characterises genetic variation among full length coding sequences of 1,000-2,000 core genes. Core genome cgMLST schemes have become an increasingly preferred approach for bacterial genotyping since whole genome sequencing is widely accessible. Analysing allelic variation across hundreds of conserved core loci leads to many unique cgSTs and this high resolution creates new challenges when comparing genomes and identifying genetic relatedness among clusters of genomes. The LIN code is a multi-position, integer-based 'barcode' that provides a stable, hierarchical, and scalable framework for genome classification. Importantly, the LIN code ensures that the classification remains stable even when new genomes are added. Each 'bin' of the LIN code corresponds to a specific cgMLST allelic mismatch threshold. Moving from left to right along the code, the bins represent decreasing numbers of allele mismatches (i.e. increasing similarity), which allows for genome comparisons that range from deep evolutionary structure to nearly identical isolates. We implemented a 10-bin LIN code system for M. catarrhalis. While there is no set rule for selecting these thresholds (we created a similar cgMLST scheme/LIN code system for pneumococci with 11 thresholds and for S aureus with 13 thresholds), the threshold choices were based on an analysis of the natural discontinuities in the M. catarrhalis population structure. We assessed this by examining the distribution of pairwise allelic mismatches amongst isolates, combined with clustering statistics including the Adjusted Rand Index and the Silhouette score. The first four bins were designed to capture deeper phylogenetic divisions (Figure 3): Bin 1 (1,290 mismatches) distinguishes the two major SS and SR lineages; Bin 2 (832 mismatches), flanks the second peak in the allelic mismatch distribution, separating genomes with 63.1–97.8% divergence; Bin 3 (281	lines 70-87, 302-318, 515-518
---	---	--------------------------------------

mismatches) captures intermediate genomic structure and aligns with an increasing Silhouette score (St = 0.72); and Bin 4 (85 mismatches) corresponds to the highest clustering quality (St = 0.75), defining finer population units.

The remaining six bins use more stringent thresholds (20, 8, 4, 2, 1, and 0 mismatches) and allow high-resolution discrimination suitable for short-term epidemiological surveillance and fine-scale genomic comparisons. These are somewhat arbitrary but reasonable choices, based on the data distribution and a desire to include bins that discriminate genomes with high resolution.

The hierarchical nature of LIN codes, presented in a single, fixed barcode for each genome, provides an easy and efficient way to represent genetic relatedness at varying level of resolution. For example, if two isolates have LIN code 4 designations of 0_2_1_1 and 0_2_1_2, respectively, we can immediately infer that both belong to SR lineage (shared '0' in the leftmost LIN code 1 bin). This is not possible using traditional STs.

While lineage assignment might be inferred from 16S rRNA analysis, this will be challenging, given that our study revealed previously unreported 16S variants. Furthermore, the LIN code reveals that these genomes differ by more than 85 but no more than 281 out of 1,319 core gene alleles, providing a quick and interpretable measure of genetic distance. The scalable design makes LIN codes applicable to a broad range of research and public health questions, including population structure analysis, outbreak investigations, longitudinal carriage studies, and transmission studies.

LIN code assignments are made via the submission of genome data to PubMLST. Users upload genome FASTA file(s), and the database curator assesses genome quality and correct species designation (via rMLST). Alleles for MLST and cgMLST are performed using the autotagger followed by manual curation. LIN code assignments are made using the curated cgMLST data, comparing the new data to all existing cgMLST and LIN code data. A LIN code is assigned to a cgST by matching it to all known LIN-encoded cgSTs: if an exact match is found then the same LIN code is assigned, but if not, then a novel LIN code is created and the bin for the corresponding threshold is incremented.

	We have added more details to several sections of the manuscript.	
4. What's missing most from this work is a meaningful exploration of the functional significance of the genomic differences identified between SR and SS lineages. Yes, differences in virulence-associated genes and antimicrobial resistance determinants are noted, but without connecting these to clinical outcomes or ecological adaptations, it's hard to appreciate their biological relevance. The analysis of virulence factors, particularly UspA1 and UspA2 structural variations, feels superficial without exploration of how these variations might affect protein function.	We agree with the reviewer that interpreting genomic differences in the context of clinical outcomes and ecological adaptation is key to understanding the biology of this organism. This is of course not a simple task, especially when clinical and other metadata for most of the non-DCHS genomes is lacking. The tools we have developed, and the findings of our study, will contribute to the next research projects with well-defined research questions and experimental testing. A recent review (ref 40) made the same point, i.e. that associations between genotype and phenotype are inconsistent and underexplored, and in particular the authors made a plea for an optimal M. catarrhalis typing scheme, which our study provides: https://www.microbiologyresearch.org/content/journal/micro/10.1099/mic.0.000523#R60. It is still unclear whether genetic variation in virulence genes contributes to differences in disease potential or colonisation. Most existing studies (e.g. ref 41) lack whole-genome data and have not examined lineage-specific variation in a population context. A few studies that incorporated limited WGS data found no consistent association between SR/SS lineages and clinical outcome (e.g ref 18). For example, in the context of COPD, one study found that M. catarrhalis strains exhibited substantial diversity within and between hosts and that no single strain type is specifically associated with exacerbation (ref: George et al, 2018). Ecological adaptation is also an underexplored area and the investigation of adaptation to distinct ecological niches would require the compilation of a different dataset to that used here. UspA: while experimental validation is beyond the scope of the current study, our comparative approach extends beyond simple gene presence/absence, which has been the primary focus of most prior genomic studies (e.g. ref 18). Specifically, we analysed structural motif variation in UspA1 and UspA2 between SR and SS lineages, offering new insights into potential functional divergence. Most experimental studies of UspA have investigated SR (but not SS) lineage isolates and the findings were not examined in a broader genomic or phylogenetic context.	lines 348-387

	To our knowledge, no previous study has systematically correlated UspA structural features including motif architecture and repeat variation with SR/SS lineage classification across a large clinical dataset. Our study fills this gap by leveraging whole-genome sequencing across 1,913 M. catarrhalis genomes, revealing clear lineage-specific patterns in UspA1/UspA2 presence, structure, and diversity. While our findings suggest that genetic variation may contribute to different protein function, experimental studies are needed to confirm whether these structural features impact protein expression, ligand binding, or serum resistance. We have revised the discussion to address these points.	
5. Finally, the pangenome analysis lacks depth. The authors report an α value of 0.9 but don't sufficiently contextualize what this means for M. catarrhalis evolutionary dynamics or how it compares to related species.	We have revised the text to clarify that although the pangenome is classified as open ($\alpha = 0.9$), it expands slowly, indicating limited gene acquisition. We now compare this to Neisseria meningitidis and Streptococcus pneumoniae, which also show slow but persistent pangenome growth. These dynamics suggest that M. catarrhalis retains enough genomic flexibility to respond to selective pressures such as antibiotic exposure or immune evasion, while maintaining a conserved, host-adapted genome structure.	lines 177-181
6. Is this the right repo at all? The description mentions: Pneumococcal Genome Library cgMLST Typing Scheme.	We have edited the GitHub page to make it clearer where to find the relevant code: https://github.com/brueggemann-lab/pgl_cgmlst_2024	
Reviewer #2		
Comment	Response	Location
1. Comparative Advantages of cgMLST and LIN Code Typing Methods: Have other researchers published similar studies utilizing the cgMLST and LIN code typing methodologies? If so, what are the distinctive advantages of your approach compared to existing methods?	Yes, we and others have developed cgMLST and LIN code systems for several major pathogens, and these genotyping approaches are implemented in PubMLST: Klebsiella pneumoniae (ref 52) Streptococcus pneumoniae (ref 26) Staphylococcus aureus (https://doi.org/10.1101/2025.03.29.646111), Haemophilus influenzae (https://doi.org/10.1099/mgen.0.001281) Neisseria gonorrhoeae (https://doi.org/10.1101/2025.03.28.646058). Regarding the advantages of the cgMLST/LIN code approach, please see our related comments and	

	changes in response to similar queries by Reviewer 1 (Q3).	
2.Hierarchical Interpretation of LIN Code Classification: The study establishes LIN code classifications based on allele mismatch rates and correlates LIN code 1, 2, and 3 with ribotype, clonal complex (CC), and sequence type (ST), respectively. While this demonstrates consistency with established typing systems in resolution, further clarification is needed regarding the phylogenetic or taxonomic hierarchy represented by LIN codes. Specifically, what deeper biological or evolutionary implications (e.g., population structure, ancestral relationships, or functional divergence) do these hierarchical levels reflect beyond mere technical concordance?	Please see our related comments and changes detailed above in response to similar queries by Reviewer 1 (Q3).	
3.Clinical and Functional Validation of Lineage-Specific Traits: (i) The manuscript highlights differences in virulence and antibiotic resistance profiles between the two lineages. Are there publicly available clinical datasets (e.g., disease severity, treatment outcomes) that support associations between these lineages and observed phenotypic variations? (ii) Regarding the proposed virulence determinants (UspA structural variation and novel LOS serotypes), the authors acknowledge	(i) To the best of our knowledge, there are no publicly available M. catarrhalis genomic datasets that include detailed clinical metadata such as disease severity, treatment outcome, or host immune status. As noted in our response to Reviewer 1 (Q4), most publicly available genomes lack associated clinical and epidemiological metadata, which limits our ability to assess correlations between lineage-specific traits and clinical outcomes. We have clarified this limitation in the revised manuscript and emphasised the need for future studies and data to address these important questions. (ii) We expanded the discussion to highlight the importance of experimental studies to better understand virulence determinants. We also added the importance of biochemical and binding assays to investigate the function of the novel LOS.	lines 348-387, 393-405

the need for experimental validation. Could the authors elaborate on specific hypotheses or mechanistic studies (e.g., animal models, gene knockout/complementation, or phenotypic assays) planned to confirm the functional significance of these genetic differences?		
--	--	--

Reviewer #3

Dear Authors,

General comment: Congratulations on your research! The topic is of great importance, especially in this context of modern investigation the population structure of *M. catarrhalis* in relation to virulence.

THE MANUSCRIPT „Understanding the population structure of *Moraxella catarrhalis* using core genome multilocus sequence typing (cgMLST) and a life identification number (LIN) code classification system” is well written in accordance with the guidelines of the journal. The title is accurate and relevant. The paper has an informative abstract. All figures and tables are necessary and understandable. The names of organisms are used appropriately. The data were systematically analyzed. The study presents relevant information for the field of two divergent SR and SS *M. catarrhalis* lineages with distinct evolutionary trajectories using whole-genome analyses of nearly 2 000 genomes from 12 countries over a long period of time. The discussion and conclusions are correct and reflect the evidence provided in the paper with a few necessary additions. The references were accurate. This work supports the discovery with the focus on the population genetics and its relationship with the virulence.

We thank the Reviewer for his/her kind comments and enthusiasm about this work.

Comment	Response	Location
The authors said that “LOS serotype distribution varied between the SR and SS lineages”. I would like to ask the authors if they have made a comparison between their results obtained with SS an SR M. catarrhalis and association between more virulent strains and LOS B?	We appreciate the reviewer’s comment. While LOS B was detected exclusively in SR genomes, the absence of associated disease/clinical severity metadata in our dataset prevents us from assessing whether this LOS type is linked to more severe disease or increased virulence. If higher virulence is interpreted as carrying more virulence-associated genes, our analysis showed that UspA1 and UspA2 were nearly ubiquitous across both SR and SS lineages. Although we observed structural motif differences in UspA between lineages, without functional validation or clinical correlation, it is difficult to predict whether any specific variant confers increased pathogenic potential. As such, we have avoided making direct claims about the virulence of specific lineages or sequence types and have noted in the discussion that future studies combining genomic, phenotypic, and clinical data will be necessary to explore these questions in depth.	lines 348-387, 393-405

And also following the discovery of ST and CC by cgMLST and evidence that SS genomes were larger than SR genomes and carry more genes, can it be said whether there are any ST or CC associated with higher virulence?	Similar to the point raised above, in the absence of linked clinical or phenotypic data, it is difficult to reliably associate any ST or CC with higher virulence.	
---	---	--